# EMO-5: A high-resolution multi-variable gridded meteorological data set for Europe

Vera Thiemig[1], Goncalo Nuno Gomes[1], Jon Olav Skøien[1], Markus Ziese[2], Armin Rauthe-Schöch[2], Elke Rustemeier[2], Kira Rehfeldt[2], Jakub Walawender[2,5], Christine Kolbe[2,5], Damien Pichon[3], Christoph Schweim[4], Peter Salamon[1]

[1]European Commission, Joint Research Centre, Ispra, 21027, Italy
[2]Global Precipitation Climatology Centre, Deutscher Wetterdienst, Offenbach, 63067, Germany
[3]Kisters France SAS, Rueil-Malmaison, 92500, France
[4]Kisters AG, Aachen, 52076, Germany
[5]Faculty of Geography, Philipps University of Marburg, Marburg, 35032, Germany

*Correspondence to*: Peter Salamon (peter.salamon@ec.europa.eu)

**Abstract.** In this paper we present EMO-5[1], a European high-resolution, (sub-)daily, multi-variable meteorological data set built on historical and real-time observations obtained by integrating data from 18,964 ground weather stations, four high-resolution regional observational grids (i.e. CombiPrecip, ZAMG - INCA, EURO4M-APGD and CarpatClim) as well as one global reanalysis (ERA-Interim/Land). EMO-5 includes at daily resolution: total precipitation, temperatures (minimum and maximum), wind speed, solar radiation and water vapour pressure. In addition, EMO-5 also makes available 6-hourly precipitation and mean temperature. The raw observations from the ground weather stations underwent a set of quality controls, before SPHEREMAP and Yamamoto interpolation methods were applied in order to estimate for each 5x5 km grid cell the variable value and its affiliated uncertainty, respectively. The quality of the EMO-5 precipitation data was evaluated through (1) comparison with two regional high resolution data sets (i.e. seNorge2 and seNorge2018), (2) analysis of 15 heavy precipitation events, and (3) examination of the interpolation uncertainty. Results show that EMO-5 successfully captured 80% of the heavy precipitation events, and that it is of comparable quality to a regional high resolution data set. The availability of the uncertainty fields increases the transparency of the data set and hence the possible usage. EMO-5 (version 1) covers the time period from 1990 to 2019, with a near real-time release of the latest gridded observations foreseen with version 2. As a product of Copernicus, the EU's Earth observation programme, EMO-5 dataset is free and open, and can be accessed at https://doi.org/10.2905/0BD84BE4-CEC8-4180-97A6-8B3ADAAC4D26 (Thiemig et al., 2021).

---

[1] EMO stands for "European Meteorological Observations", whereas the 5 denotes the spatial resolution of 5 km.

**How to cite**. Thiemig, V., Gomes, G.N., Skøien, J.O., Ziese, M., Rauthe-Schöch, A., Rustemeier, E., Rehfeldt, K., Walawender, J., Kolbe, C., Pichon, D., Schweim, C. and Salamon, P.: EMO-5: A high-resolution multi-variable gridded meteorological data set for Europe, Earth Syst. Sci. Data Discuss., 2022.

## 1 Introduction

Many environmental models rely heavily on the availability of meteorological data. Factors like the accessibility, quality, spatio-temporal coverage as well as spatio-temporal resolution of those meteorological data influence and ultimately determine their modelling capacity. This is further intensified for environmental applications that are running operationally and require

quality-controlled, multi-variable meteorological data in near-real time. One prominent example that relies heavily on good quality meteorological input data provided in near real-time is the European Flood Awareness System (EFAS), which is part of the Emergency Management Service (EMS) of Copernicus, the EU's Earth observation programme. EFAS provides a flood monitoring and forecast service for riverine and flash floods across the whole of Europe. The forecasts of EFAS are calculated using the semi-distributed hydrological rainfall-runoff model LISFLOOD (https://ec-jrc.github.io/lisflood/) which relies

heavily on quality-controlled, (sub-)daily meteorological information on precipitation, temperature, wind speed, solar radiation and water vapour pressure for a) model calibration and validation as well as for b) the computation of initial conditions during the operational running. In 2006, during the set-up phase of EFAS for pre-operational running there was an imminent need for a pan-European, quality-controlled, high-resolution, multi-variable, near real-time as well as historical meteorological data set. However, despite the existence of a fairly good coverage network of in situ stations across Europe, at that time, there existed

no overarching service that collected in near real-time all those meteorological in situ data across the entire European domain. For this reason, the Joint Research Centre started in 2006 with the collection, quality control and gridding of real-time and historic (from 1970) meteorological data across Europe and neighbouring regions, a service that became in 2012 known as the Copernicus EMS Meteorological Data Collection Centre (MDCC).

The CEMS MDCC runs around the clock and produces daily near real-time meteorological grids which are used in the operational running of not only EFAS, but also by two other major CEMS services, namely the European Forest Fire Information System (EFFIS; https://effis.jrc.ec.europa.eu/; San-Miguel, J. et al., 2019) and the European Drought Observatory (EDO; https://edo.jrc.ec.europa.eu/; Spinoni et al., 2016; Cammalleri, C. et al., 2020). At the same time, the MDCC collects also historical data in an offline mode and feeds those into the MDCC data collection for the production of historical

meteorological grids. This is important as historical data (produced with the same method) are indispensable for the calibration of the various models and indicators. While the service produces daily updated grids (EMO-5 operational grids), we have re-run our archive from 1990-2019 to produce a new long-term dataset, which we refer to as version 1 of the EMO-5. EMO-5 (version 1) comprises daily 5x5 km grids for six variables - **precipitation**, **minimum and maximum air temperature**, **wind speed**, **solar radiation** and **water vapour pressure** - and additional 6-hourly grids for precipitation and **mean air**

**temperature**. The underlying data comes from a total of 26 data providers that contributed to the MDCC data collection, by sharing data from a total of about 18,694 in situ stations across Europe as well as five gridded data sets over selected areas. The gridded data sets have been added to the MDCC data collection, in order to improve the quality of the resulting meteorological grids by increasing the information density over selected areas, mainly data scarce areas and areas with complex topography.

Throughout the last decade, many other observational meteorological data grids emerged, such as: E-OBS (Haylock et al., 2008; Cornes et al., 2018) for the whole of Europe; seNorge2 and seNorge2018 for Norway (Lussana et al., 2018); SPREAD for Spain (Serrano-Notivoli et al., 2017); ZAMG-INCA for Austria (Haiden et al., 2011); CombiPrecip for Switzerland (Sideris et al., 2014); CarpatClim for Hungary, Serbia, Romania, Ukraine, Slovakia, Poland, Czech Republic and Croatia (Antolović et al., 2013; Spinoni et al., 2015); EURO4M-APGD for the European Alps and adjacent flatland regions (Isotta et al. 2014); SAFRAN for France, Spain and Tunisia (Quintana-Seguí et al., 2008; Vidal et al., 2010; Quintana-Seguí et al., 2017; Tramblay et al., 2019). However, despite the availability of these data sets and their immense value each of them holds, EMO-5 represents a uniquely valuable resource due to a combination of its pan-European coverage, near real-time production, high temporal (6-hourly and daily) and spatial (5x5 km) resolution, large amount of input data (18,694 in situ stations and five high resolution regional observational grids), seven different variables (i.e. precipitation, minimum, maximum, and mean air temperature, wind speed, solar radiation and water vapour pressure), and a substantially long historical data record (from 1 January 1990). These characteristics combine to make EMO-5, to our knowledge, the most complete gridded multi-variable observational near-real time meteorological data set covering the whole of Europe (and peripheral areas).

The aim of this paper is to present EMO-5 (version 1) and its potential usage, by providing an insight into its data sources, the applied methods and the quality of the resulting data set. In this paper, the evaluation of the resulting grid quality is focused mainly on the gridded precipitation data, as a) precipitation is the most crucial driver for hydrological modelling (our main focus) and b) the minimum and maximum temperature of EMO-5 have been already investigated by Lavaysse et al. (2018) (EMO-5 was referred to as LisFlood in their study). However, we invite the scientific community to expand the evaluation exercise beyond EMO-5 gridded precipitation to other variables and other validation approaches including through various environmental applications.

Users should be aware that EMO-5 is prepared principally for near real-time rather than climatological applications. EMO-5 (version 1 as well as the operational grids) is an operational data set based on the maximum amount of quality-controlled information available at any given time. Environmental applications, especially those with real-time, high spatial resolution or multi-variable needs are likely to benefit from this data set. Hence, by making the EMO-5 (version 1) data publicly available, we aim to support many other environmental applications and services that would benefit from using those data, such as e.g. hydrological, agricultural or other environmental applications.

The remainder of this paper is organised as outlined in the following. The source data are described in Section 2. The entire workflow of the grid creation, including the quality control criteria applied during the data collection and an evaluation of various interpolation methods, are described in Section 3. Data access information are given in Section 4. An evaluation of the grid quality for precipitation is described in Section 5, and finally, some conclusions are presented in Section 6, followed by a future outlook.

## 2 Input data


The meteorological data for EMO-5 (operational and version 1) come from 26 data providers (i.e. 21 station data providers, plus 5 gridded data set providers), mostly being national meteorological services, and a few international or regional bodies. Table 1 shows the full list of data providers contributing to EMO-5.

[insert Table 1 here]


The MDCC collects historical (from 1970) and real-time observations obtained from 18,694 in situ meteorological stations across Europe, and an additional 13,394 meteorological stations globally. It should be noted that while some stations measure multiple variables, others measure only one or two. Furthermore, some stations provided data for the entire period from 1970, while others provided data only for a limited time-period. The MDCC collects 13 different meteorological variables:

precipitation, 2m air temperature (i.e. measured at 2 metres above ground), daily minimum and maximum 2m air temperature, 10m wind speed (i.e. measured at 10 metres above ground), 10m wind direction, cloud cover, water vapour pressure, solar radiation, sunshine duration, relative air humidity, evaporation, and dew point temperature. Of these 13 variables, the following seven are used for EMO-5 (version 1): precipitation; minimum, maximum and mean air temperatures; wind speed; water vapour pressure; and solar radiation.


Besides the actual in situ meteorological data, the station metadata including latitude, longitude, elevation, and (where available) instrument specifications, are collected. Data collection times depend on the data type (real-time or historical), the data availability and the chosen data transfer method. In general, real-time data are collected in a 24/7 mode as soon as they become available, whereas historical data are collected mostly on an annual basis. Regarding temporal resolution, for all

variables except precipitation, the highest resolution (from 10 minutes upwards) is preferred as it provides the possibility to aggregate the data to multiple levels, which is useful both in its own right and for the quality control, as it increases its robustness. For precipitation on the other hand, the longest reported totals are used to calculate the daily and 6-hourly totals. Naturally, the number of variables per station varies, as does also the completeness of the data record per station.

Ingesting information from high-resolution regional observational grids has been shown to improve the quality of the final grids, in particular over areas with complex topography and / or low station density (Gampe and Ludwig, 2017). For this reason, the MDCC collects also data from four high-resolution regional gridded data sets (namely CombiPrecip, ZAMG-INCA, EURO4M-APGD and CarpatClim) as well as ERA-Interim/Land over areas with low station densities or with complex topography (see Fig. 1). For all regional observational gridded data sets having a spatial resolution higher than the one currently

used in the European Flood Awareness System (implying all but CarpatClim), a regular subset of grid points with horizontal resolution of around 10x10 km was selected for integration into the MDCC data collection. The CarpatClim and ERA-Interim/Land data sets were imported at their original resolution. Each selected grid point is treated as a virtual station in the database. In total 10,632 virtual stations were added to the database for EMO-5. The main characteristics of the five input meteorological data grids underlying EMO-5 are summarized in Table 2:

[insert Table 2 here]

For each of the seven EMO-5 meteorological variables, the location (and hence density) of the input data, as well as the record length per station and the number of input stations over time (1990-2019), are shown in Fig. 1 to 3 respectively. As can be expected, the number of available stations for each variable and grid realisation is not constant over time or space. Jumps in

the data coverage are caused by the integration of historical gridded data sets with fixed start and (partially) end dates, as well as the integration of historical data from data providers, beginning after 1990. This temporally and spatially inhomogeneous availability of stations leads to an inhomogeneous time series of grid-cell values, and therefore the EMO-5 data set is not optimised for trend (or temporal) analyses, but ideally suited as input data for a wide range of environmental and hydrological model applications.

[insert Figure 1, Figure 2 and Figure 3 here]

All data collected by the MDCC, are covered by the EEA-EUMETNET Public Duty License Agreement or the EEA non-EUMETNET Partner license Agreement, which means those data are share-able between the Copernicus services.

**3 Methodology**

**3.1 Quality control on input data**

All of the data collected by the MDCC undergo an automatic quality control procedure, irrespective of whether or not they have already been checked by their specific data provider. The quality control is implemented based on five types of data validation rules:

1)  Availability: Check if value is present and time-stamp correct.
       2)  Monthly statistics: Check each value against statistical monthly data.

3) Cross-validation: Check each value against values from other parameters.

4) Minimum / maximum validation: Check each value against minimum / maximum thresholds.

5) Rate of change validation: Check the rate of change between two values against maximum thresholds.


The exact specification of the validation rules depends on the variable type as well as on the aggregation level, as summarized in Table 3.

[insert Table 3 here]

A data value is flagged as "missing" if the value is not available, and as "suspect" if the time stamp has been corrected (Rule 1). Observations are expected at time stamps according to WMO regulations (Manual on Codes, 2013), Offsets up to 29 minutes are corrected. Some countries do not report 6hourly, 12hourly or daily precipitations totals at 00UTC, 06UTC, 12UTC and 18UTC, but at 03UTC, 09UTC, 15UTC and 21UTC. Those data are shifted by three hours forward in time to match the expected reporting times. The error due to the time shift is smaller than disaggregate the data to equally distributed hourly

totals, which are accumulated to 6hourly and daily totals afterwards. It is also flagged as "suspect" if it fails the validation against the monthly statistics (rule 2) or the cross-validation (rule 3). For the monthly statistics, the mean monthly minimum and maximum for each station is calculated with the available data and updated with an annual cycle. A data value is flagged as "rejected" if it falls outside of the defined minimum / maximum range (rule 4) or if the threshold for the maximum rate of change between two values has been exceeded (rule 5).

**3.2 Choosing an optimal spatial interpolation scheme**

As the primary usage of EMO-5 is to support an operational flood forecasting service, the data need to be delivered in a timely manner and at a good level of quality, while the operational framework needs to be robust and easy maintainable. Three spatial interpolation schemes - Inverse Distance Weighting (IDW), Modified SPHEREMAP, and Ordinary Kriging - were compared and evaluated in terms of reliability, specifically regarding uncertainty and computational cost. The three interpolation schemes

are briefly described in Table 4.

[insert Table 4 here]

The quality of each of the interpolation schemes was derived through a leave-one-out cross-validation. This means that for

each iteration of the interpolated field, one station was left out and then later on compared with its interpolated value. This was done for around 4000 randomly chosen stations evenly distributed over high and low density station areas, and those pairs of interpolated and real observations were used to compute the uncertainty estimates. A similar approach was applied by Hofstra et al. (2008) for the ECA&D data set. The three tested interpolation schemes used the same setting, as far as possible. These identical settings were: at least four and at maximum ten stations were used to compute the grid cell value and no restriction

in the distance to find the nearest four neighbouring stations. The initial search radius to find the nearest stations was calculated as described in Shepard, 1968. Further scheme dependent settings are mentioned in Table 4.

Three different measures of errors were calculated, which are the Mean Error (ME), Mean Absolute Error (MAE) and Mean Squared Error (MSE), as each focus on different aspects of uncertainties.


[insert Table 5 here]

As is shown in Table 5, SPHEREMAP was found to be the best scheme regarding ME, but IDW outperforms both other
schemes regarding MAE and MSE. The variance between observed and computed value (MSE) is lowest for IDW for four out of six parameters. Only Kriging has lower variances for precipitation and vapour pressure. In addition, IDW has the lowest values for four out of six parameters regarding MAE. Here, SPHEREMAP performs a better interpolation for precipitation and vapour pressure. Regarding computation times, Kriging needed on average around 700 seconds, SPHEREMAP around 550 and IDW around 470. It should be noted that oceans were not masked to speed up the computations.

In favour of obtaining an operational framework that is maintainable, the decision was taken to choose only one interpolation scheme for all variables. The analysis of the error measures shows that none of the tested interpolation schemes outperforms the others in a consistent manner. However, working within an operational environment, the robustness of the system is crucial, and from that point of view, the automatic variogram fitting in Kriging is a stability concern that was also observed by Ntegeka et al. (2013), and hence was excluded for EMO-5. As SHPEREMAP shows the best performance for the critical parameter
precipitation, it was ultimately chosen as the interpolation scheme to generate the grids of EMO-5.

There are many methods available to estimate the uncertainty (i.e. reliability) of the gridded values, such as the leave-one-out approach, ensemble creations or the technique developed by Yamamoto (2000). Kriging itself provides an error estimation, but this depends only on the spatial distribution of the applied stations and not on the input data, therefore this error estimation
is not applicable here. As the computational time of the grids is highly relevant in order to produce the operational grids as input for emergency management applications, the technique developed by Yamamoto (2000) is used due to its low computational effort. Furthermore, this method takes into account the variability of the surrounding observations, unlike the common Kriging uncertainty that only depends on the variogram and the spatial distribution of stations. This approach was also used, for example, for the E-OBS data set (Haylock et al., 2008). Originally, it was developed for Kriging schemes, but
was adapted to the utilized modified SPHEREMAP scheme. Briefly, the method uses the interpolation weights to calculate a weighted variance between the gridded value and the input station data. It is zero if all input data are identical (e.g. areas with zero precipitation) and increases with increasing variance of the input data. This method does not need any additional information besides grid value, station values and the interpolation weights.

Uncertainty fields were calculated and analysed for each grid realisation and interpolation method. Figure 4 below shows the gridded daily precipitation data and the corresponding uncertainty fields (for 15[th] May 2014) as an example for IDW (left), SPHEREMAP (middle) and Kriging (right).

[insert Figure 4 here]

The overall patterns for precipitation look very similar for the three interpolation approaches (Fig. 4, top row), with the only

main difference being that Kriging produces smoother patterns compared with IDW and SPHEREMAP. Comparing their affiliated uncertainty patterns (Fig. 4, bottom row), SPHEREMAP shows the lowest uncertainty, and unlike the other two interpolation schemes the uncertainty signal of SPHEREMAP is geographically constrained to the regions with precipitation. Kriging shows extremely high gridding uncertainties, which we assume are due to a not well-fitted variogram.

The specific date (15[th] May 2014) is just one example that we selected from the comparison study of the interpolation methods, but it could have been any other day since the results were generally the same, to wit: 1) Kriging generated smoother variable fields than SPHEREMAP and IDW; and 2) SPHEREMAP showed overall the lowest and Kriging the largest uncertainty. Further, the interpolation performance (meaning the stability of the algorithm in a near real-time setting) was considered, as the timely availability of the gridded meteorological data must be assured in order to guarantee the smooth operation of the

flood forecasting system. Unlike for IDW or SPHEREMAP, this is an issue for Kriging as the latter would require an automatic fitting of the variogram, which might go wrong in a near real-time operation. Based on the results of the uncertainty analysis, and the consideration of the interpolation performance, SPHEREMAP was chosen as the interpolation algorithm for EMO-5.

## 3.3 Grid creation

Moving from a large collection of in situ measurements to a European high-resolution, (sub-)daily, multi-variable meteorological data set, involves several processes and decisions, which are described in the following subsections.

### 3.3.1 Selection of stations

The number of stations used during gridding varies per variable and time step. This is due to the fact that for each grid creation the stations in the EFAS meteorological database are filtered based on a number of criteria. A station passes the filtering

process for a particular variable and time step, if it fulfils all of the following criteria, in order to be used as input for gridding:

- The data coverage of aggregated precipitation readings is 100% for the entire period of the time step.
- The data coverage for all remaining variables such as temperature, wind speed, etc. is 95% for daily data and 83% for 6-hourly mean temperature.

▪    The recorded values for that time-stamp passed the data quality check (see Section 3.1), meaning that they were flagged as "good" or "suspect" (data with quality "rejected" are excluded from gridding).

For precipitation, not all station records that fulfil the above criteria are used in the interpolation. This is due to the fact that over time, there was an increasing number of stations that were reported redundantly by different data providers (e.g. SYNOP

and national data), albeit sometimes with slightly different values or slightly different coordinates. Hence appearing as multiple station in the database, while in fact they are one station with multiple records. Not removing those duplicate records would lead to a multiple counting of the same station during the interpolation, with the result of overweighting of those stations in the grids and less reliable area mean grid-cell values. To correct this, and to assure a gradual change between stations during the interpolation, redundant records were identified through a vicinity check. Records, i.e. stations within a vicinity of 500

metres to each other were identified and merged into one record, i.e. virtual station. The coordinates of the virtual station were taken from the first station of this cluster, while the value was computed as the average of all duplicate records. This reduced the total number of records used per grid realisation by an average of 3.4%. Figure 1 shows the number of stations used during the grid creation.

Another filter that was implemented for precipitation on the level of the interpolation was a distance filter for ERA-

Interim/Land data. The reason for this is that ERA-Interim/Land was used with the intention to fill the spatial gaps where no observations were available. However, with time the station density in those areas increased and therefore the need to integrate ERA-Interim/Land stations decreased. For this reason, all ERA-Interim/Land values that were less than 100 km away from a valid in situ station measurement were disregarded. Figure 1 shows how many of the 1402 existing virtual ERA-Interim stations were actually used for gridding.

**3.3.2 Aggregation or reference period**

Figure 5 shows the aggregation (i.e. reference period) for the different variables. Daily precipitation is accumulated from the current day at 06:00:01 until the next day at 6:00:00. (Note: "current day" = "reference date of grid"). In accordance with the WMO guidelines (WMO No.306, 2019), the minimum night-time temperature measured between 18:00:01 on the previous day until 06:00:00 on the current day is taken as the minimum temperature of the current day; and the maximum day-time

temperature measured between 06:00:01 and 18:00:00 is used as the maximum temperature of the current day. All the other daily parameters such as wind speed, solar radiation and vapour pressure are averaged over 00:00:01 of the current day to 00:00:00 of the next day. For 6-hourly temperature, this depends on the data provider: if only 6-hourly instantaneous temperature readings are delivered, then those are used, otherwise temperature is averaged for 00:00:01-06:00:00, 06:00:01-12:00:00, 12:00:01-18:00:00, and 18:00:01-24:00 (see Section 3.3.5). The same time intervals are used to aggregate 6-hourly

precipitation reported at 06:00, 12:00, 18:00 and 24:00 respectively. Note that the time-stamp of the EMO-5 grids refers to the end of the reference period, meaning that, for instance, the daily precipitation of 22.02.2021 covers the time period between 21.02.2021 06:00:01 and 22.02.2021 06:00:00.

[insert Figure 5 here]

### 3.3.3 Land-sea mask

A land-sea mask is used to exclude sea surfaces from the gridding procedure, as EMO-5 originates from the need for near real-time information on observed meteorological conditions over land surface areas. The land-sea-mask is not used to define stations to be excluded from gridding.

### 3.3.4 Implications of altitude for temperature and water vapour pressure

In areas with strong orographic changes, neighbouring stations are likely to be at various altitudes, and hence the
representativeness of their measurement for neighbouring areas is limited depending on the altitude change in the surrounding area (due to adiabatic processes). If station values were blindly used during the interpolation, an error would be introduced. This can be easily avoided by considering elevation information while interpolating. As SPHEREMAP does not take any auxiliary information into account, all recorded temperature and water vapour pressure values are brought to sea-level before interpolation. This is done based on altitude information obtained from a 1x1 km Digital Elevation Model (Arnal et al., 2019),
through accounting for the adiabatic change of 0.006 Kelvin per height metre for temperature and 0.00025 hPa / metre for water vapour pressure. The interpolation runs at sea-level and afterwards temperature and water vapour pressure information are brought back to the mean elevation of the respective 5x5 km grid cell by taking the parameter-specific correction factors into account.

### 3.3.5 Mean temperature

Where six hourly temperature averages are available, these have been used both for the six hourly data set and for the daily averages, aggregating the four 6-hourly observations. However, for many stations we did not have sub-daily temperature observations, particularly for the data from the 1990s. There are also many stations in the data set that have frequent observations between 06:00 and 18:00, enough for the 6-hourly daytime, but not for the nighttime period. This is particularly the case for observational data received from the MARS Meteorological Database of the European Commission's Joint
Research Centre (Toreti et al., 2019). If data from other providers are not available at any station suffering from this issue, we have estimated the missing 6-hourly temperature based on the daily minimum and maximum temperatures. This was done based on the method described as the "Sin(14R-1)" method in Chow and Levermore (2007) with some small modifications as follows.

For all stations where minimum and maximum temperatures are available, we fitted sinusoidal curves after estimating at what time of day the extremes most likely occurred. Similar to Chow and Levermore (2007), we assumed that the minimum occur 1 hour before sunrise, a value we computed with the function "getSunlightTimes" of the "suncalc" package (Thieurmel and Elmarhraoui, 2019) in the R programming language. However, we restricted the earliest time of the minimum temperature to

occur not before 04:00 UTC in summer, with a correction for longitude. The latest time for the minimum was set at 10:00 UTC in winter, also corrected for longitude. While the "Sin(14R-1)" method assumes that the maximum temperature occurs at 14:00, we let the peak time range between 14:00 and 16:00 UTC, where it will be closer to 16:00 when sunset is late.

As some of the stations with missing 6-hourly data have stations with high-resolution observations nearby, we only included the estimated values for stations where the distance to a high-resolution neighbour is at least 10 km. This value was chosen by analysing the mean square error for simulated temperatures at stations where we have 6-hourly averages and a variogram analysis of the 6-hourly data series. Requiring a mean square error of around 5.5 $K^2$ roughly corresponds to a distance of about 10 km in the variogram. Beyond this distance, it is likely that the errors from simulation are smaller than the spatial interpolation error. The number of extra stations is highest for the period 1990-1995, with around 500-600 stations with estimated 6-hourly temperatures. This can be compared with the approximately 1000 stations with 6-hourly temperature, as seen in Fig. 3. The number is then reduced to approximately 50-150 stations for most of the remaining period, but increases above 200 again for 2019 (mainly because of stations in Iceland).

Figure 6 shows the available stations for three different years (i.e. 1990, 2005 and 2019), for the 00:00 and 12:00 observations. The red dots show where we have 6-hourly temperature data, whereas the blue dots show where we have been able to add simulated 6-hourly temperature data based on the minimum - maximum observations. We can see that the simulated data covers much of Europe in 1990, whereas it drops for the later years. For the most recent years, most of the extra stations are added in North-Eastern Russia, some around the Mediterranean, and a large number in Iceland, where we have access to many more stations with minimum - maximum observations than 6-hourly data. For 1990, we can also notice the rather big difference between the data from nighttime (top) and daytime (bottom), where we have 6-hourly data from many more stations during the day than during the night.

[insert Figure 6 here]

The goodness of this approximation was analysed in a simple cross-validation procedure. We picked around 800 stations that already had 6-hourly observations from the start of the period (1990). For these stations, we compared the 6-hourly observations with the simulated 6-hourly dataset based on maximum and minimum temperatures. A hexbin-plot is shown in Fig. 7, where darker colors indicate a high density of points indicate a temperature with a large number of observed and simulated values. The line shows where simulations are equal to the observations, whereas the dashed lines indicate where the simulated values are 2 degrees higher or lower than the observations. We can see that a large majority of the simulated values are within 2 degrees of the observations, although there are cases with larger deviations.

[insert Figure 7 here]

The correlation between observations and simulations range from 0.91-0.99. The simulated values are on the average unbiased, with mean and median around 0.1 degrees below the observations. However, when looking at the simulated temperatures for different times of the day, it can be noted that the method has a tendency to underestimate the average night temperature, and overestimate the afternoon temperature. The mean and median of the underestimation is 1.8 degrees for the night temperature and the overestimation 1.5 degrees for the afternoon, i.e., the simulated 6-hour periods are slightly more extreme than the observations. The root mean squared error (RMSE) for each station ranged from 1-4 degrees, with mean and median RMSE of 2.2 degrees. The difference was seen as acceptable for our purposes, and is quite similar to what was observed by e.g. Luedeling (2018).

### 3.3.6 Historical batch creation versus near real-time creation

The amount of station data available for gridding differs if these are created in near real-time or once in a batch for a given historical time period, with more stations available for the latter. This is due to the latency in data availability from climatological stations, which do not report in real-time mode but are included in historical data deliveries. The earliest station data are available and transferred into the MDCC database about 0.5 hours after the observational time, whereas all near real-time reporting stations including corrections are acquired within 5 days after the observational time. To account for the incremental filling of the database in the near real-time grid production, at every grid production cycle, grids of day-6 (i.e. six days ago) to day-2 are being produced (where "today" is defined as "day-0") and overwrite those previously produced. In this way, the latest grids always contain the highest amount of information available at any given moment.

This incremental gridding strategy is not necessary if one would like to generate grids for a historical period, as the gridding is executed once in a batch process considering all the information available in the database at that given moment. The recreation of grids is performed according to need. Once a year all grids of the previous year are reproduced to account for the delivery of historical data from the data providers. The recreation of the whole time series is much rarer, and only happens after larger changes, for example after the change of the interpolation algorithm or the resolution (spatial or temporal), or the integration of larger amounts of historical data.

The EMO-5 (version 1) dataset  has been produced for the time period from 1990 till 2019. In order to have a data set that is consistent over time, it has been reproduced in a batch process for the entire time period using the modified SPHEREMAP algorithm and also for both temporal resolutions (6-hourly and daily).

**4 Data availability**

EMO-5 (Thiemig et al., 2021) is a Copernicus product and as such free and open to everyone. It can be accessed through the Data Catalogue of the European Commission's Joint Research Centre at https://doi.org/10.2905/0BD84BE4-CEC8-4180-97A6-8B3ADAAC4D26.

The repository contains a CF-1.6 compliant NetCDF stack files for each variable, as well as a README file with detailed product specifications, which are briefly summarised in Table 6 below.

[insert Table 6 here]

**5 Evaluation**

The quality of the EMO-5 precipitation data is evaluated in this section through (1) examination of the interpolation uncertainty, (2) comparison with two regional high resolution data sets (i.e. seNorge2 and seNorge2018), and (3) analysis of 15 heavy precipitation events.

**5.1 Interpolation Uncertainty**

As interpolated grid-values represent a "best guess", it is of paramount importance to provide information on the reliability. EMO-5 therefore contains, for each variable and time step, two fields - the interpolated value and an interpolation uncertainty, created with the methods described in Section 3.2.

An example of the associated interpolation uncertainty for the interpolated precipitation totals for the extreme event number 3 (see Section 5.3 below) is shown in Fig. 8. The highest accumulated precipitation totals are around 300 mm and the estimated accumulated interpolation uncertainty is up to 100 mm. In the largest parts of the extreme event area, the estimated accumulated precipitation uncertainty is around 25 mm, which is less than 20% of the accumulated precipitation total. Obviously, the highest estimated uncertainties are not located at the highest totals, but rather at the slopes of the highest rainfall areas. It is worth mentioning that the estimated interpolation uncertainty in the precipitation-free areas is roughly zero.

[insert Figure 8 here.]

**5.2 Comparison against a regional high resolution grid**

We compared EMO-5 with two Norwegian data sets in order to assess its quality. One is the data set "seNorge2" (Lussana et al., 2018) and the other is the subsequent data set "seNorge_2018" (https://doi.org/10.5281/zenodo.2082320, Lussana, 2018b). The area covered by these two data sets consists of the Norwegian mainland and a strip of the neighbouring countries Sweden, Finland, and Russia (Lussana, 2018).

The covered area is very challenging for raster data generation with large differences in the orography, strong precipitation and large small-scale precipitation differences (e.g. due to the orography). Both seNorge data sets are based on station data from the Norwegian Climate Database (data.met.no) and outside Norway on the European Climate Assessment data set (Klein Tank et al., 2002). For the seNorge2 data set the raw data are used while for the seNorge2018 data set an undercatch correction is applied, which has a large impact especially in the winter months when snow is detected. The data are on a regular grid with 1-km grid spacing for the periods 1957-2015 for seNorge2 and 1957-2017 for seNorge2018. Both data sets have daily values from 6 UTC to 6 UTC. An extra interpolation based on nearest neighbour is used to identify the precipitation days per grid point, or otherwise to set the grid values to zero. The interpolation procedure for the precipitation values for both data sets is based on an optimal interpolation procedure with a background field. The seNorge2 data set uses the station data as a background field, while seNorge2018 uses the monthly values of the ERA-Interim analysis with 2.5 degrees resolution. For the actual interpolation, a cascade of decreasing scales is used and the station height is taken into account. The differences are mainly seen in mountainous regions with sparse station coverage. Since the optimal interpolation procedure requires a normal distribution, a Box-Cox transformation is applied to the seNorge2018 data set (Lussana et al., 2019).

For comparison with EMO-5, the period 1990-2015 which is common to all data sets, is used. The day definition (from 6 UTC to 6 UTC) is the same for all three data sets. To obtain the same spatial projection, the seNorge data sets were reshaped to the 5-km resolution of EMO-5. For this purpose, the bilinear interpolation method was applied, which uses a weighted average of the 4 nearest grid points. The study area of this analysis can be seen in Fig. 9.

[insert Figure 9 here.]

The differences in annual precipitation, as well as the differences in the distributions of seasonal precipitation, and the distributions of extremes represented by the precipitation indices adopted from the CCl/WCRP/JCOMM Expert Team on Climate Change Detection and Indices (ETCCDI) set by Peterson et al. (2001, Appendix A), are presented below.

As can be seen in Fig. 9, the data have a similar spatial structure, with seNorge2018 showing heavier precipitation. This is more clearly visible in the direct comparison in Fig. 10.

[insert Figure 10 here.]

Figure 10 shows that EMO-5 and seNorge2 have very similar magnitudes of values. The differences between the two data sets are very small-scale, especially in the southwest and northeast. These could indicate orographic effects, as the seNorge interpolation methods include the elevation as a parameter. Large structures can be found in the centre of the area, which could indicate fundamental data differences. In comparison with seNorge2018, the high precipitation values of seNorge2 are evident again. This is not surprising due to the undercatch correction in this data set. However, the presence of two conspicuous blue-coloured areas shows that EMO-5 is characterized by higher precipitation than seNorge2018. The seasonal differences are

listed in Table 7 (differences in the values shown in Table 7 and Fig. 10 are due to the slightly different procedure to derive the maximum).

[insert Table 7 here.]

The average annual mean precipitation of EMO-5 is 2% higher compared to the one of seNorge2, and 17% lower compared to seNorge2018. In the average maximum precipitation per grid, EMO-5 shows 106% of the seNorge2 precipitation and 86% of the seNorge2018 precipitation. EMO-5 is thus substantially closer to seNorge2 in mean grid precipitation than the two seNorge data sets are to each other (see Table 7).

[insert Figure 11 here.]

The comparison of the seasonal precipitation in Fig. 11(a) shows that the individual months are very different. There are very large differences especially with respect to intense precipitation. These are distinctly higher in EMO-5 than in the seNorge data sets, an exception being autumn. In this case, the distributions of seNorge2018 and EMO-5 agree very well. Large EMO-5 data values are rare, only 1523 values (or 0.06%) are above 1500mm, and 212 values ($9^{10-5}$%) are above 2000mm.  The

reason is probably data errors, as these values can be located in a few small locations in the study area. For the benefit of visual comparability, the highest 0.01% of the values are omitted in Fig. 11.

The probability distribution of the smaller values, on the other hand, shows very good agreement with seNorge2, while seNorge2018 shows larger values especially in spring and winter. The comparison of the two seNorge data sets shows

significantly higher values for seNorge2018, especially in spring and winter. In summer, the seNorge data sets are very similar. The latter finding can be explained very well by the fact that the undercatch correction has less impact in summer. The results look more diverse when considering the extremes on a daily basis (see Fig. 11(b)). The maximum extreme value indices are calculated over the entire time period for each grid point. Again, as in Fig. 11(a), the highest 0.01% of the respective values are not shown. This concerns for example the high values of the Consecutive Dry Days (CDD) - up to 1400 days without

precipitation - in the EMO-5 data set. These unrealistic values can be limited to a small region in the northeast. The high values for the Consecutive Wet Days (CWD) can be attributed to two points in the centre and one in the south of the study area. Otherwise, the extra interpolation for the Yes-No decision for precipitation days in the seNorge data sets could have an influence on the distributions of the CDD and CWD indices. The Simple Daily Intensity Index (SDII) shows similar distributions as the totals in Fig. 11(a). SeNorge2 and EMO-5 agree very well, while seNorge2018 shows much higher values.

The number of precipitation days above 10 mm and above 20 mm (not shown) shows a good agreement between seNorge2 and EMO-5, similar to SDII, with a slightly better agreement at r10. The maximum 5-day totals show a strikingly different picture. Here EMO-5 agrees very well with seNorge2018, while seNorge2 shows significantly lower values.

In summary, the agreement between the seNorge2 and EMO-5 data sets is very good, despite the very different interpolation methods. The influence of the undercatch correction used in the seNorge_2018 data set is substantially larger. In particular, the lower precipitation values seem to match very well in seNorge2 and EMO-5. This is also evident in the extreme value indices based on low values, especially r10. The ratio of the annual values clearly shows the different representation of the orographic structure due to the orography-dependent interpolation in the seNorge data sets. The large differences in the extremes with very few values are mainly due to isolated data errors in the EMO-5 data. However, the differences in the isolated grid points do not appear to be caused by single extremely high daily values, as these would otherwise show in indices such as the Maximum Consecutive 5-Day Precipitation (RX5day). Instead, smaller but longer-lasting errors seem to cause these discrepancies. These then affect statistics that use all values over a period of time, such as the seasonal totals or indices like CWD and CDD.

## 5.3 Heavy precipitation events

As well as the timeliness of EMO-5, also the capability to capture heavy precipitation events is of great importance for EFAS. In this Section, we evaluate semi-quantitatively the ability of EMO-5 to capture heavy precipitation events. We first randomly selected from www.floodlist.com, 15 flood events that were caused by heavy precipitation. Affiliated precipitation information was then extracted and verified through visual inspection of the respective EMO-5 image.

Figure 12 shows for each of the 15 selected flood events the related precipitation information as published on Floodlist (on the left-hand side) and as seen in the EMO-5 data set (all four maps). The information regarding the observed precipitation as published on Floodlist ranges from a qualitative indication, such as "heavy precipitation between those dates" (Events 2, 3, 4, 11 and 14) to precise quantitative statements such as "x amount of precipitation between those dates" (Events 1, 5-10, 12, 13 and 15). When the original data source of the observed precipitation was mentioned on Floodlist, then it is also cited here. For all the other cases the value shall just be used as an indication.

Regarding EMO-5, we present here the sum of daily and 6-hourly precipitation maps covering the dates as specified on Floodlist, as well as the maximum daily and 6-hourly precipitation values within that time frame. In order to cover a full calendar day it was necessary to sum up 2 daily precipitation maps. This is due to the fact that the aggregation or reference period of the daily precipitation maps is not in line with a calendar day, but covers the period from 06:00:01 of the previous day until 06:00:00 of the current day (see Section 3.3.2). Hence, there is always one more daily map than the number of calendar days to cover. This was not necessary for the 6-hourly maps, as four 6-hourly maps cover precisely one calendar day. This explains also why in Fig. 12 the sum of daily precipitation is generally higher than the sum of the 6-hourly precipitation. This should not be a problem, as the aim of this comparison is to evaluate the performance of the daily and 6-hourly precipitation against the reported information, and not to compare the 6-hourly with the daily totals.

For 13 out of the 15 selected events, EMO-5 shows greater precipitation amounts over the event duration ranging between 119 mm (Event 5) to 350 mm (Event 9), with 270 mm and 255 mm being the maximum observed precipitation within a daily and 6-hourly period respectively. The only event clearly missed is Event 6, where hardly any precipitation was captured by EMO-5. This event was caused by a cloud burst, and hence was a very local extreme event. It is likely that it was missed as no in situ station exists directly at the specific location in EMO-5.

For the nine events with concrete specifications on the precipitation amounts, EMO-5 captures the precipitation amounts of three events (Events 5, 8 and 15) in accordance with the media reports, overestimates the amount for one event (Event 9), and underestimates the reported amounts for five events (Events 6, 7, 10, 12 and 13) . Three out of the five underestimated precipitation events (Events 6, 10 and 12) were also caused by cloud bursts, and hence were short and high intensity events that are difficult to reproduce in a temporally aggregated and spatially interpolated data set, especially if the full amount of precipitation was not captured by any in situ station, due to the positioning of the stations.

[insert Figure 12 here]

## 6 Conclusions and future work

EMO-5 is a European high-resolution (5x5 km), (sub-)daily, multi-variable, multi-decadal meteorological data set based on quality-controlled observations coming from almost 30,000 stations (18,964 in situ and 10,632 virtual stations) across Europe, and is produced in near real-time. The EMO-5 (version 1) covers precipitation, temperature (minimum and maximum), wind speed, solar radiation and water vapour pressure all at a daily resolution, and in addition 6-hourly for precipitation and mean temperature. Version 1 covers the time period from 1990 onwards and is freely available. In this paper we have provided insight into the source data, the applied methods and the quality assessment of EMO-5 version 1 (the latter just for precipitation).

EMO-5 grids are produced by means of a modified SPHEREMAP algorithms, which is a geometric scheme taken the distance, clustering and gradient into account. The decision was made after a comparison of three interpolation schemes. An algorithm was developed to identify stations provided independently by different data providers and replace them by a merged station. Each EMO-5 grid realisation is accompanied by a corresponding estimate of the interpolation uncertainty, which is based on the approach by Yamamoto (2000). This takes the difference between the gridded value and the station data as well as the interpolation weights into account. Therefore, the estimated uncertainty at each grid cell differs from day to day. The highest uncertainties of the gridded data are at steep gradients between regions with high and low precipitation totals, reflecting the uncertainty in the estimation of the spatial extension of such an event.

The quality of the precipitation product was evaluated through comparison against a regional high-resolution data set (seNorge2 and seNorge2018) over Norway as well as against 15 media reports of extreme precipitation events that caused flooding across Europe.

The comparison of EMO-5 (version 1) against two regional high resolution data sets over Norway has shown that EMO-5 and seNorge2 are more similar than both Norwegian data sets are to each other. The good agreement is despite their very different interpolation methods, and is particularly evident for lower precipitation values and mean annual means, while differences can be detected in the seasonality and the extreme values. The latter issue is suspected to be caused by longer-lasting data errors of very few isolated stations used in the EMO-5 data set. The reason for seNorge2018 being substantially different to both EMO-5 and seNorge2, is likely due to the undercatch correction that is applied only to seNorge2018. It is recommended to broaden this comparison to other high resolution data sets before drawing general conclusions. However, the comparison done here has shown that EMO-5 can be comparable in terms of quality with a regional high-resolution data set.

The semi-quantitative evaluation of EMO-5's capacity to capture heavy precipitation events has shown that in 80% of the cases, EMO-5 captures the events qualitatively, meaning that the EMO-5 grids covering the event show large amounts of precipitation, which is important in particular for applications such as flood modelling, that rely heavily on the forcing data to be able to capture those events. As expected for gridded data sets, even if EMO-5 mostly captures heavy precipitation events, it tends to underestimate the observed precipitation amount at stations. This is not a surprising finding as especially convective events are often short and local. This makes it very difficult to capture the maximum precipitation amount of those events, unless an in situ station is in the direct location of the event and captures it.

The presented EMO-5 (version 1) is the result of an operational real-time service, which utilizes for every grid realisation the maximum amount of valid information available. This is very different to climate data sets, which need to be of sufficient length, consistency, and continuity to determine climate variability and climate change. EMO-5 favours the maximum amount of available information over long-term product homogeneity and is therefore not suitable for climate studies i.e. trend analysis. However, EMO-5 is suitable for environmental application which do not have this homogeneity requirement, but value a data set that uses the maximum amount of information available for each grid realisation.

Further, there might be some reservations towards using the EMO-5 minimum night-time and maximum day-time temperature as the daily minimum and maximum temperature respectively. To recall, the daily minimum and maximum temperature within EMO-5 are calculated following the WMO guideline (WMO-No. 306), which assumes the minimum temperature to occur between 18:00 and 06:00, and the maximum temperature between 06:00 and 18:00. However, particularly in the winter half season and in the higher latitudes, the daily maxima or minima temperature occur sometimes outside these windows. For this reason, some data sets, such as E-OBS, calculate the daily minimum and maximum temperature over the full 24 hour period. Lavaysse et al. (2018) who investigated the minimum and maximum temperature of EMO-5 (referred to as LisFlood in their paper) and E-OBS with regards to their suitability in the frame of temperature extremes in Europe (heat- and cold waves),

came to the conclusion that the two observational datasets showed only minor differences in heat and cold waves occurrences and intensities, which according to Lavaysse is probably due to the good agreement in representing both, the minimum and maximum temperature.

Looking forward, as part of the operational Copernicus EMS, the number of stations (historical and near real-time) that are used for gridding in EMO-5 will be continuously increased, through adding new data providers and the integration of new, high-resolution regional observational grids, where available. In addition, to improve the current quality control framework, new data validation rules, such as spatial comparison with neighbouring stations or additional statistical checks, will be

implemented. Finally, as it is foreseen to increase the spatial resolution of EFAS from the current 5 km grid to a 1 arc minute grid (approximately equal to 1.8 km at the equator), also EMO-5 will increase the spatial resolution in its next version.

Lastly, the CEMS Meteorological Data Collection Centre publishes every year an annual report on the CEMS meteorological data collection, with updated information on e.g. data providers and provision, database, post-processing, improvements to the

590 data flow and post-processing and gap analysis. All reports can be found on the EFAS website, while last years report is referenced here: Rehfeldt, et al., 2021.

**Author contribution**

MZ, ARS, KR, JPW, CK, DP are the core team of the CEMS Meteorological Data Collection Centre (MDCC), which is overall

managed by CS. Their work is the foundation for the creation of EMO-5 as they are running the daily operation of the historical and near real-time data collection, the design and operation of the data validation and the data gridding method. GG supports the operation of this service at all levels and has created the actual EMO-5 data set. KR, CK, ARS and MZ have implemented and evaluated the interpolation method used for EMO-5, which MZ has documented in section 3.2 and section 5.1. PS is the project coordinator of the flood forecast and monitoring component of CEMS supported by VT, and as such have guided the

CEMS Meteorological Data Collection Centre from the start, including all its evolutions since and has sparked the idea of EMO-5. ER has single-handed lead the comparison of EMO-5 against the seNorge data sets as she documented in section 5.2. JOS has selected, implemented and documented the methodology for the computation of the mean temperature (section 3.3.5) and provided scientific support on the entire methodology. VT was in charge of the overall coordination of the manuscript, carried out the analysis in section 5.3, and is main writer of the manuscript, with valuable support on technical details from all

co-authors.

**Competing interests**

The authors declare that they have no conflict of interest.

**Acknowledgements**

Our sincere gratitude goes to all the data providers who shared and continue sharing their (station) data with the CEMS Meteorological Data Collection Centre (CEMS MDCC). A full list can be found in the supplementary material. The CEMS MDCC is funded by the Copernicus program, and runs as part of the Copernicus Emergency Management Service (CEMS). We would furthermore like to thank the Copernicus in situ cross coordination action implemented by the European Environmental Agency for its support in setting up in situ data licenses for meteorological data; and lastly, Niall McCormick (JRC) for English proofreading and Christel Prudhomme (ECMWF), Carlo Buontempo (C3S) and Andrea Toreti (JRC MARS) for reviewing.

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

## Figures

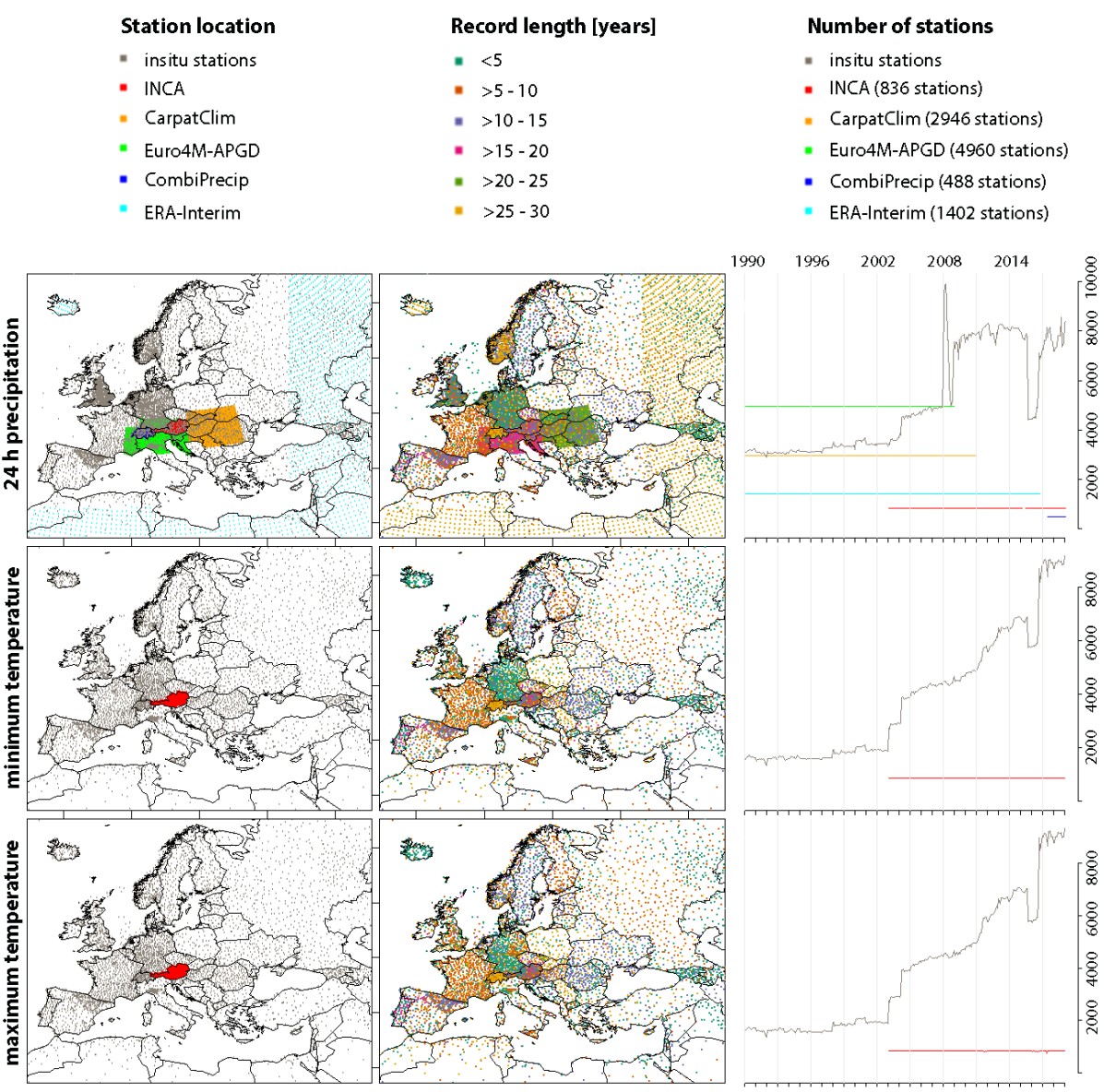

Figure 1: Input data for daily EMO-5 precipitation and minimum and maximum temperature grids in terms of station locations, 740    record lengths and number of stations used.

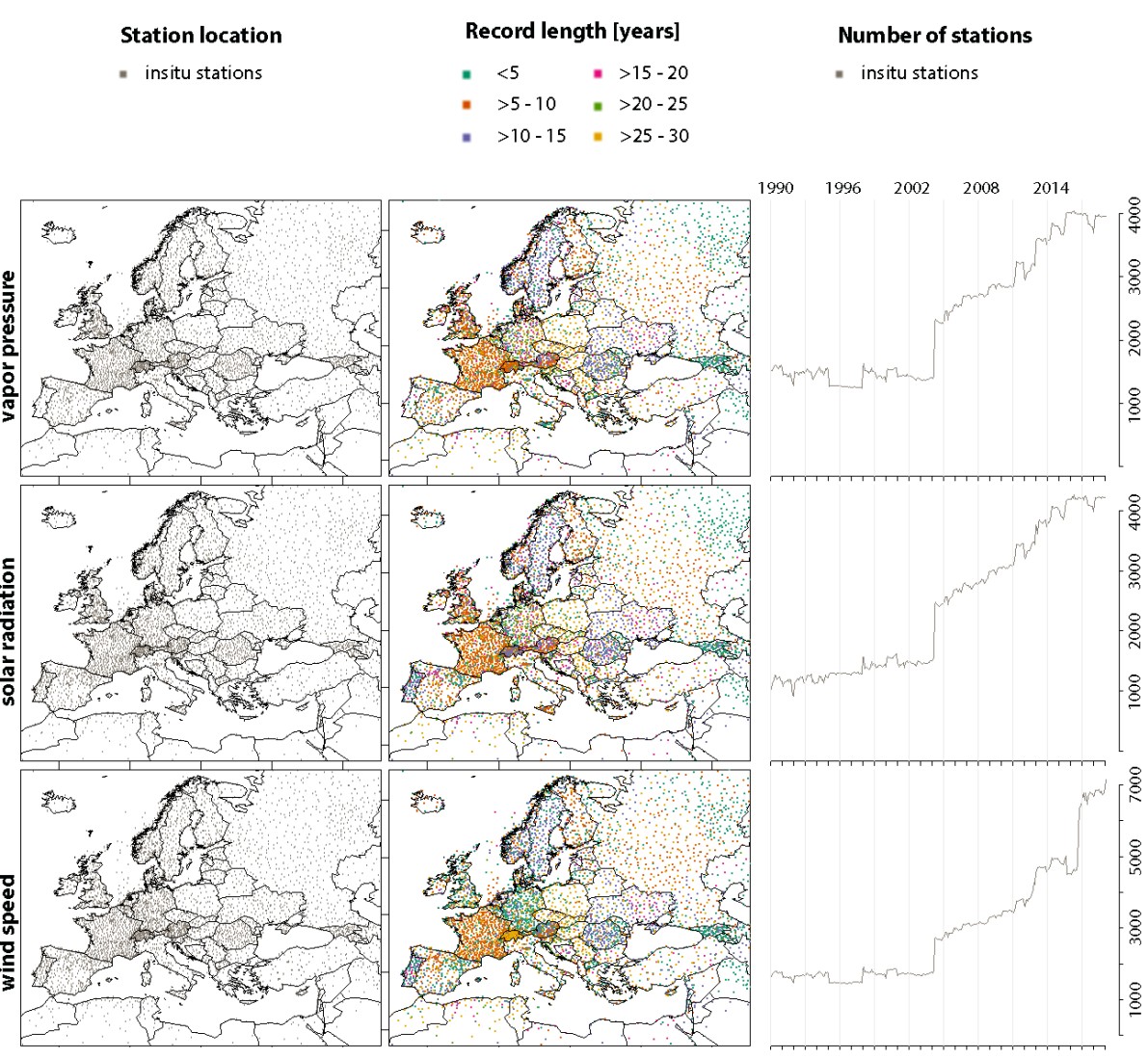

**Figure 2: Input data for daily EMO-5 vapour pressure, solar radiation and wind speed grids in terms of station locations, record lengths and number of stations used.**

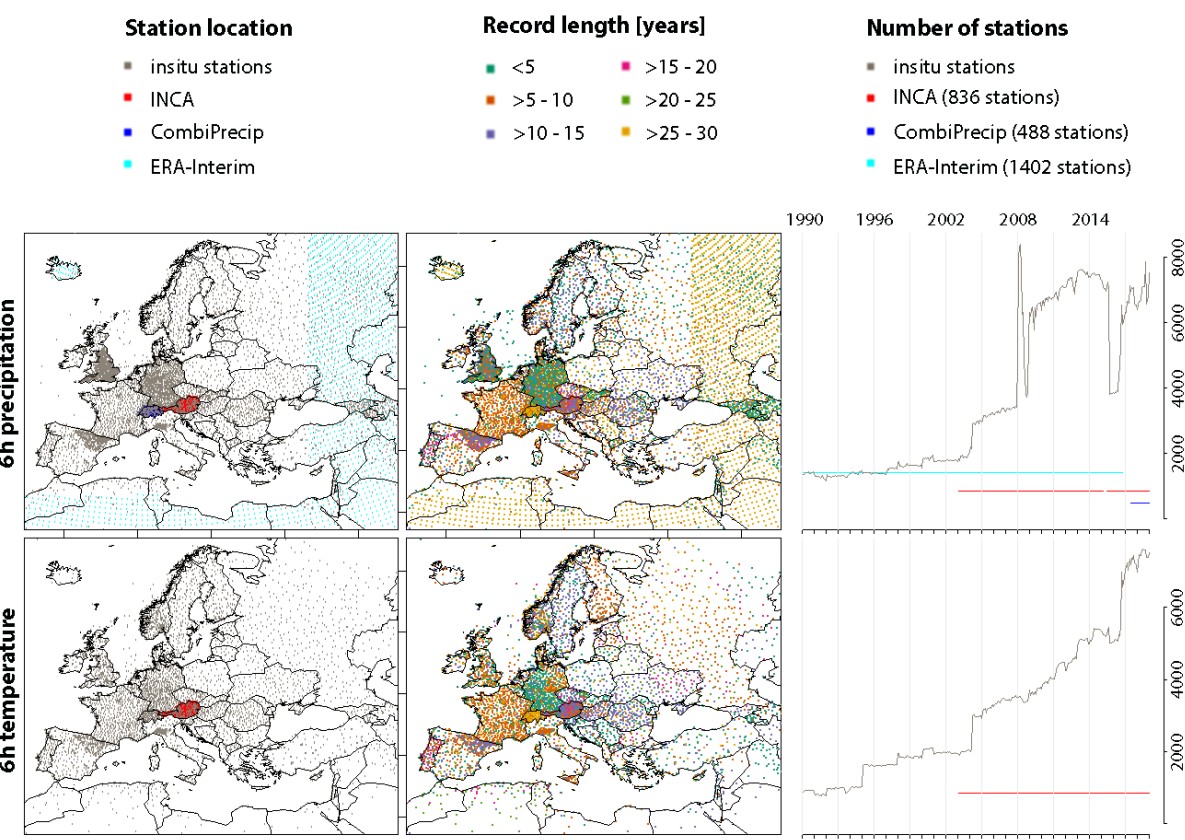

Figure 3: Input data for 6-hourly EMO-5 precipitation and temperature grids in terms of station locations, record lengths and number of stations used.

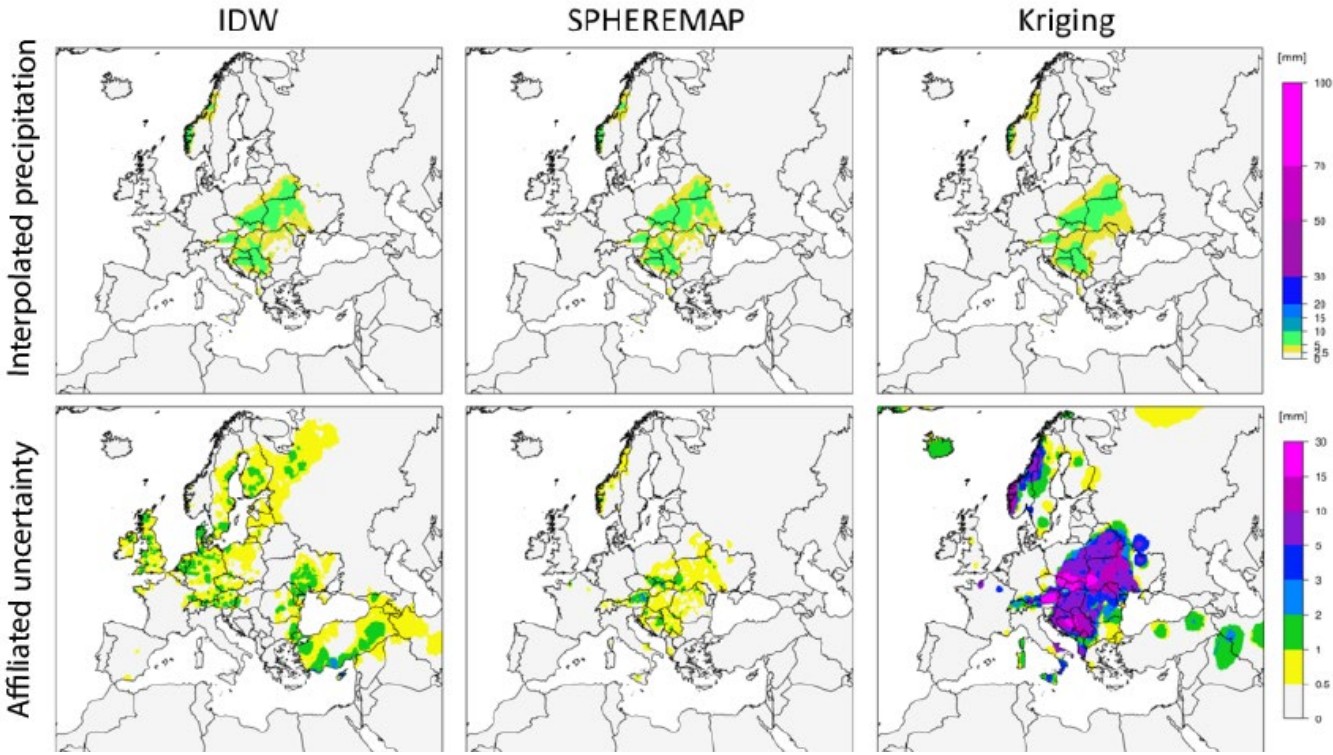

**Figure 4: Gridded daily precipitation data (top row) and the corresponding uncertainty fields (bottom row) for the 15.05.2014 as an example for IDW (left), SPHEREMAP (middle) and Kriging (right).**

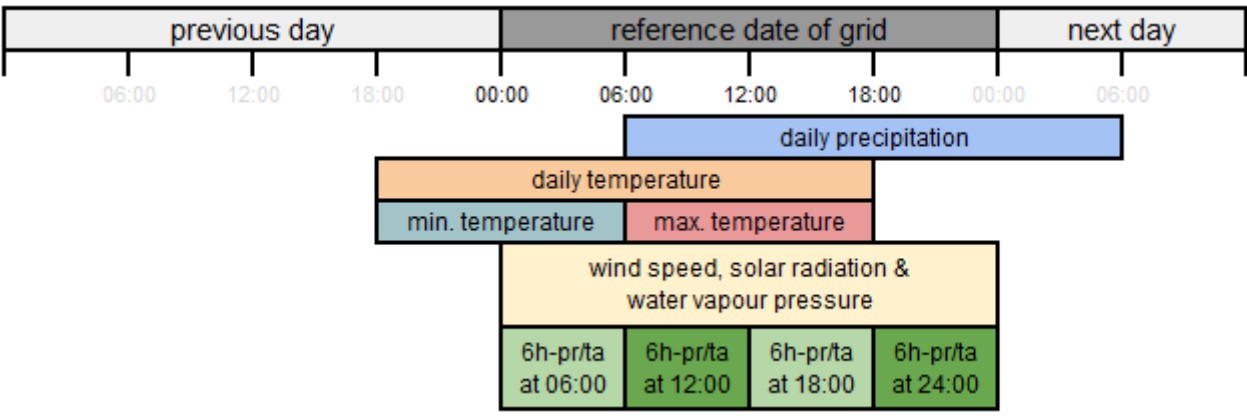

**Figure 5: Aggregation i.e. reference period of the EMO-5 grids per variable.**


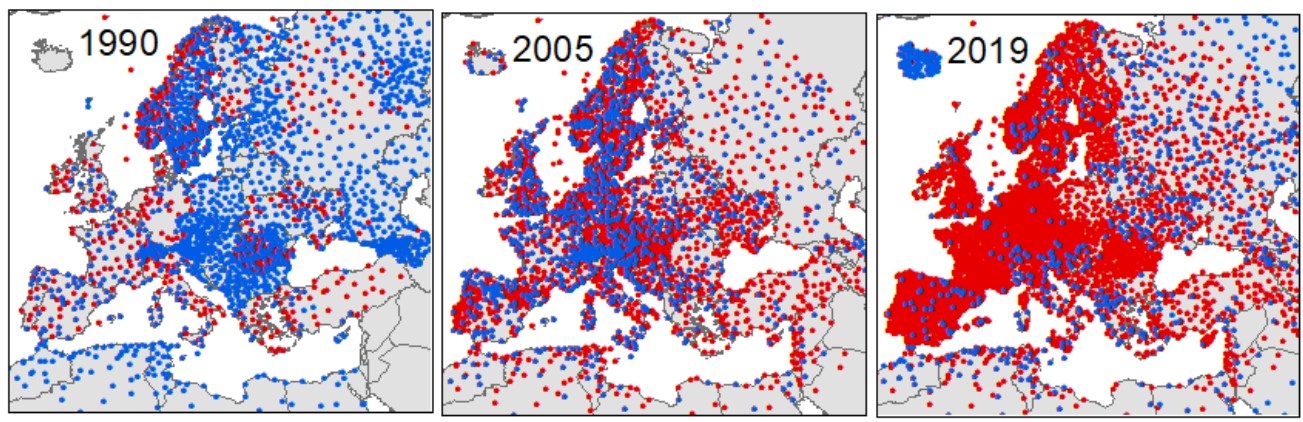

Figure 6: Coverage of existing and simulated stations used for the 6-hourly temperature fields for 1990, 2005 and 2019.

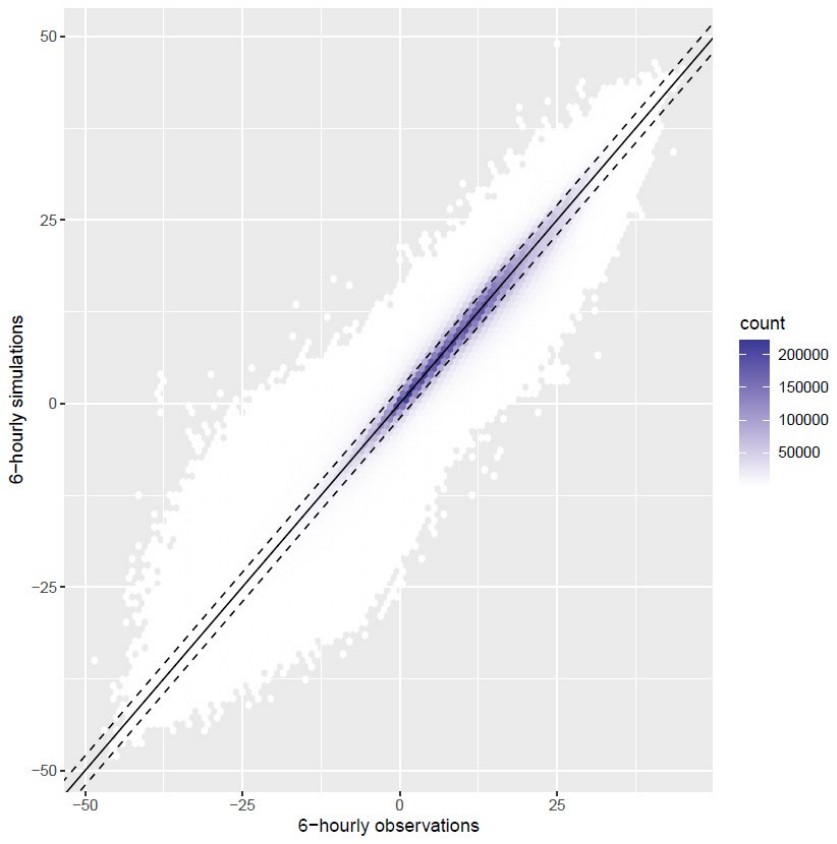

**Figure 7: Comparison of observed and simulated 6-hourly temperatures.**

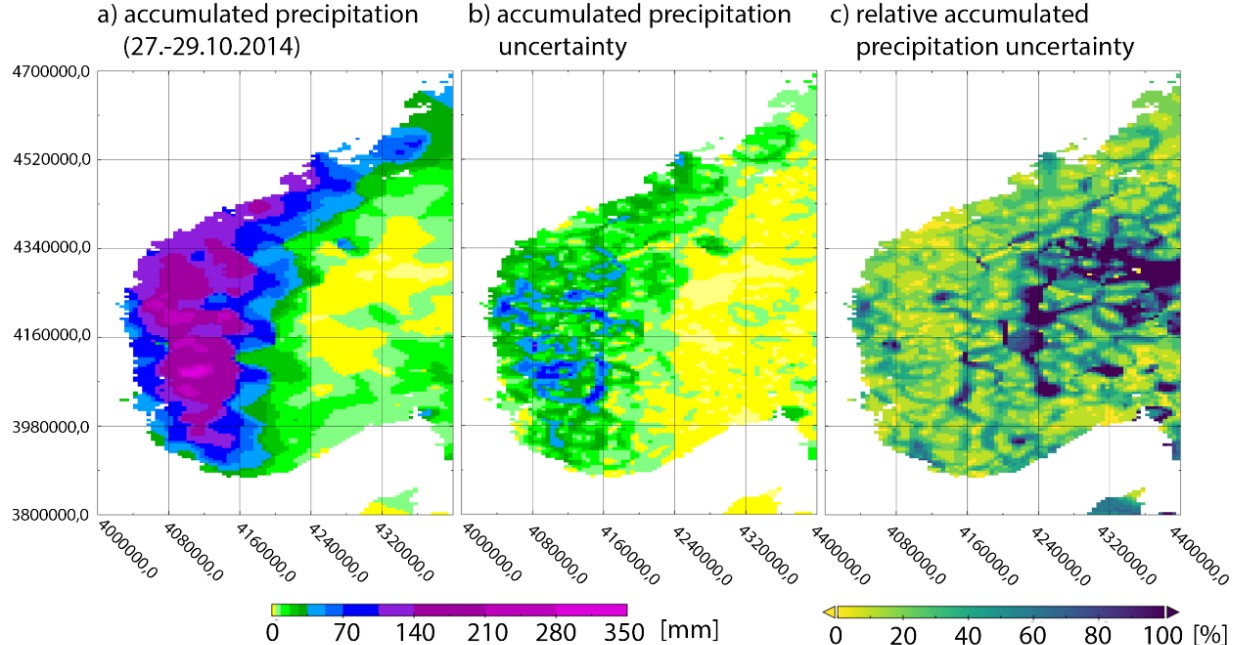

**Figure 8: Example of interpolated precipitation totals and associated uncertainty for extreme event No. 3 in Section 5.3. (A) Accumulates precipitation totals from 2014/10/27 to 2014/10/29 (three days), (B) accumulated uncertainty of the interpolated precipitation totals and (C) the percentage of the accumulated uncertainty in relation to the accumulated precipitation totals.**

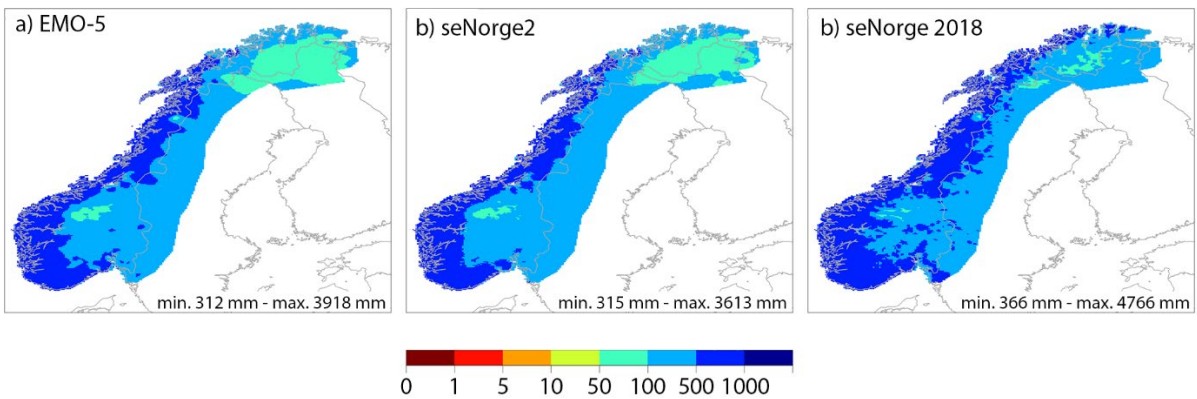

**Figure 9: Mean annual precipitation [mm] for the (a) EMO-5, (b) seNorge2, and (c) seNorge 2018 data sets for the joint period 1990-2015. seNorge data are bi-linearly interpolated to the coarser EMO-5 resolution. EMO-5 was cropped to the same spatial extent as the seNorge data sets.**

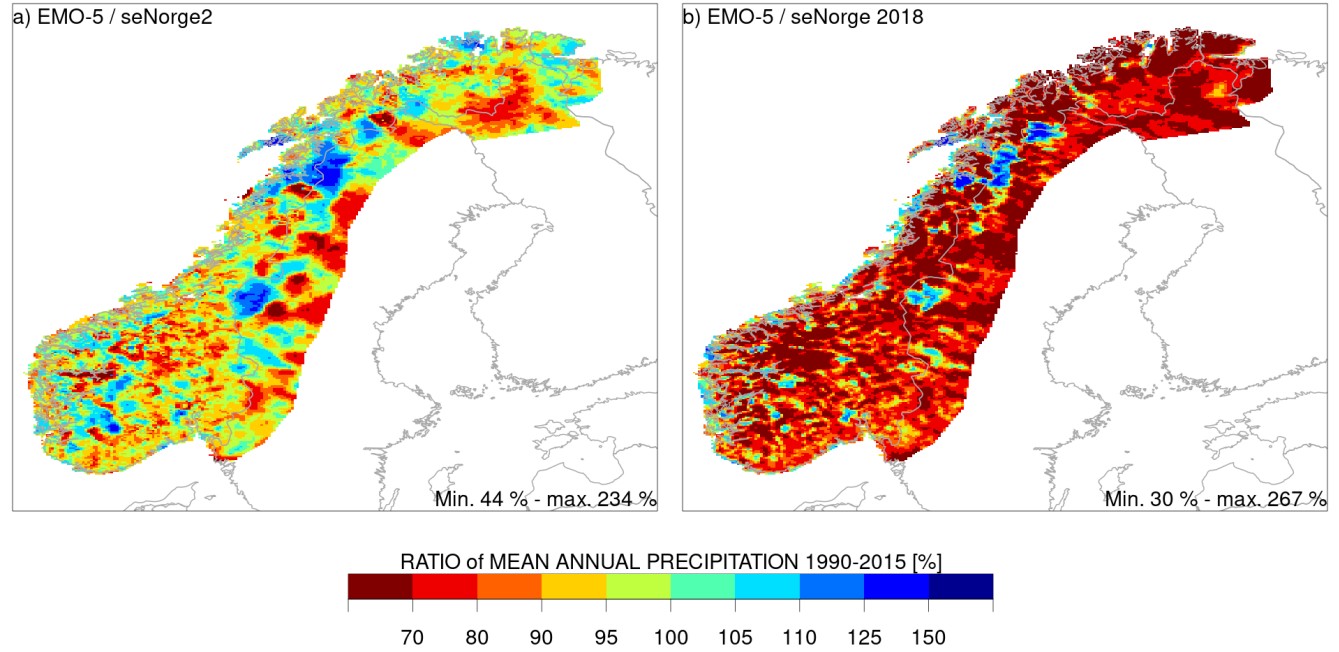

**Figure 10: Based on Figure 8 data. Ratio of seNorge2 (a) and seNorge 2018 (b) to EMO- 5 [%].**

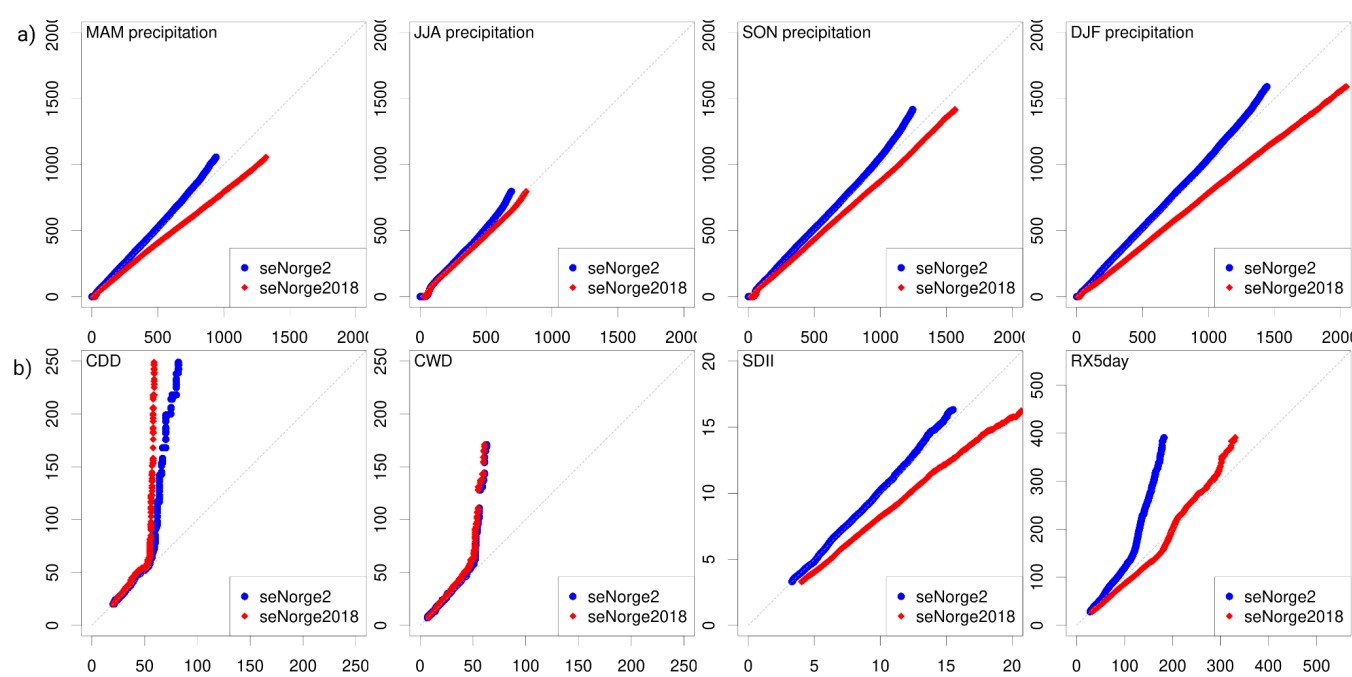

**Figure 11: Quantile plot of a) seasonal precipitation, where each point represents the sum of a grid value over 3 months and b)**
**extreme value indices (ETCCDI) values, but these correspond to the maximum of the entire time series. Each plot shows seNorge2**


(●) and seNorge2018 (♦) values on the x-axis and EMO-5 values on the y-axis in mm. For a better visual overview, the highest 0.01% of the data have not been shown.

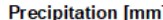

**Precipitation [mm]**

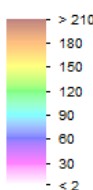

> 210
180
150
120
90
60
30
< 2

| | sum of daily maps | sum of 6-hourly maps | max of daily maps | max of 6-hourly maps |
|---|---|---|---|---|
| **1. Italy: Region of Grosseto**<br><br>About 150 mm of rainfall between 06 and 07 October 2013. | max = 89 mm | max = 41 mm | max = 47 mm | max = 16 mm |
| **2. Sweden: Halland, Värm-land & Västa**<br><br>Long periods of heavy rain from 18 to 22 August 2014. | max = 221 mm | max = 203 mm | max = 129 mm | max = 78 mm |
| **3. Norway: Sgn, Fjordane**<br><br>2 Days of extreme rainfall between 27 and 29 October 2014 | max = 320 mm | max = 229 mm | max = 150 mm | max = 55 mm |
| **4. Albania: near Tirana**<br><br>Heavy rainfall on 22 November 2015. | max = 163 mm | max = 142 mm | max = 88  mm | max = 78 mm |
| **5. North Macedonia: Skopje**<br><br>93 mm in 3 h between 6 and 7 August 2016 | max = 119 mm | max = 87 mm | max = 91  mm | max = 54 mm |

**6. Italy: Sicily, province of Agrigento (city of Licata)**

160 mm of rainfall between 19 and 20 November 2016.

**7. Spain: Murcia**

Up to 400 mm between 16 and 19 December 2016.

Source: Agencia Estatal de Meteorología

**8. Germany: Berlin and surrounding**

150 mm rain between 29 and 30 June 2017.

Source: Deutscher Wetterdienst

**9. Croatia: Zadar**

Torrential rainfall of 280 mm between 11 and 12 September 2017.

Source: Croatian Meteorological and Hydrological Service

**10. Italy: Livorno**

Up to 250 mm of rainfall in 2 hours on 10 September 2017.

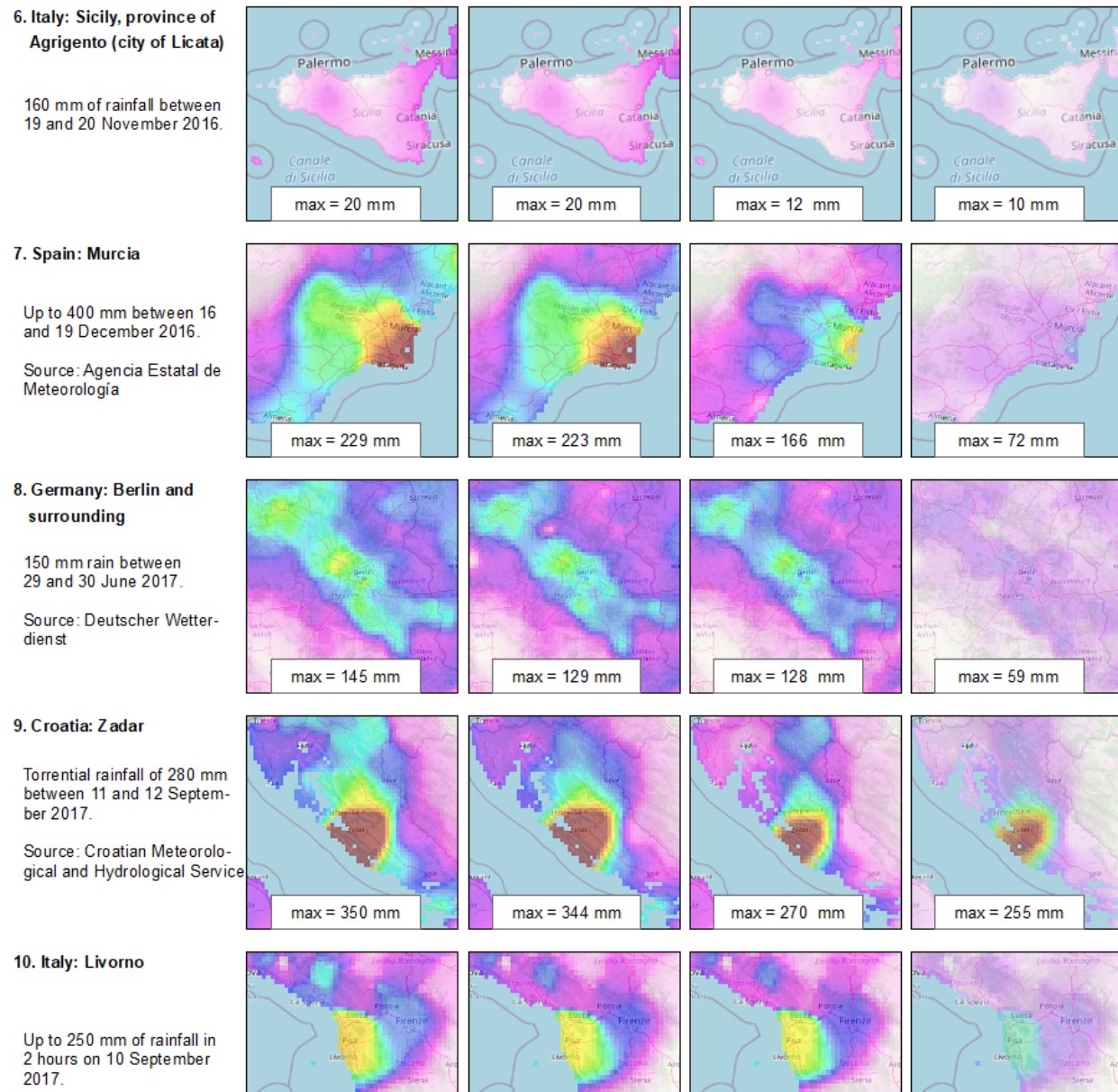


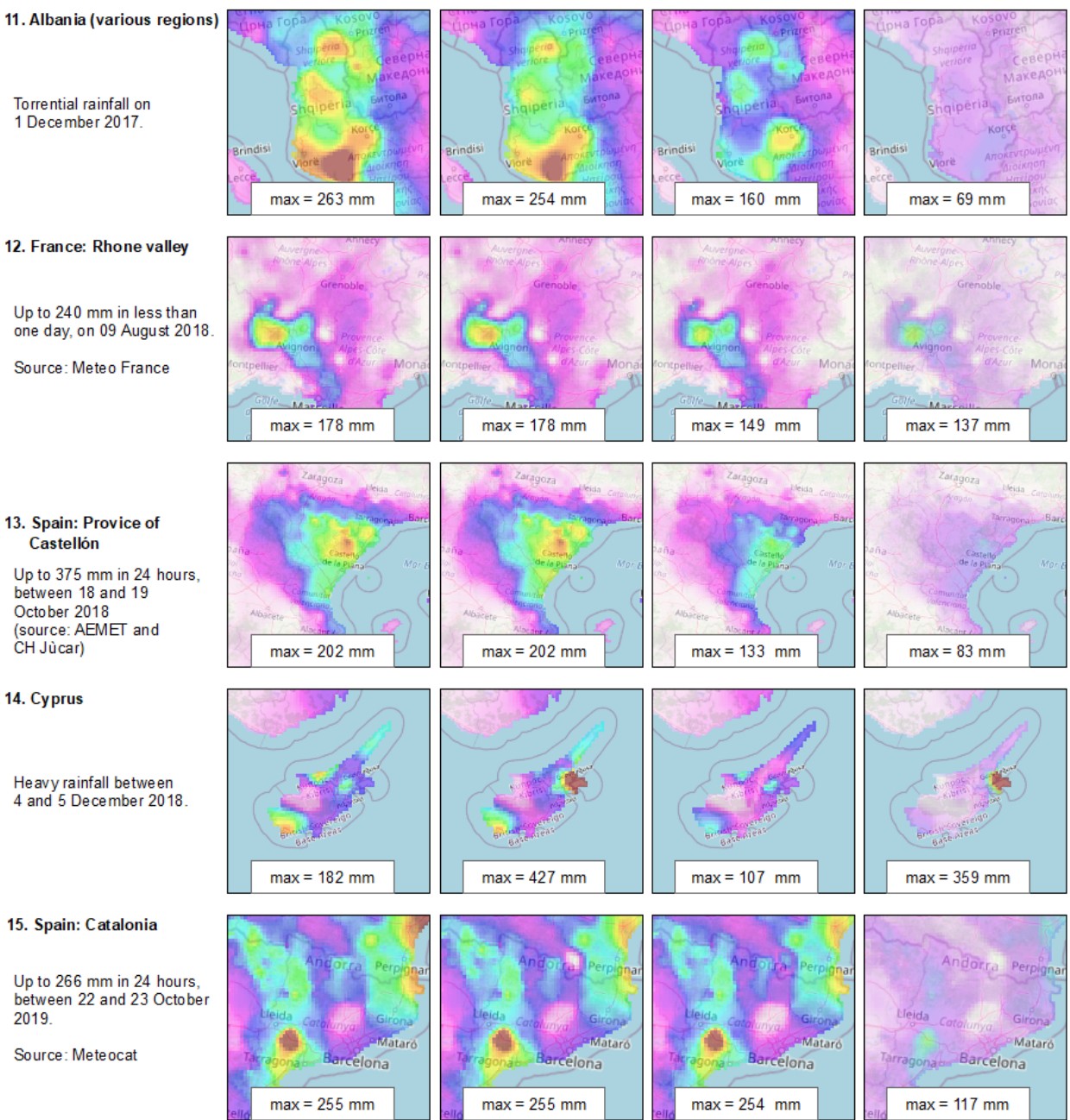

**Figure 12: Semi-qualitative evaluation of 15 high intensity precipitation events through comparing information reported on FloodList (on left) with the footage of the 6-hourly and daily precipitation grids (sums and maximum) (base map from © OpenStreetMap contributors 2020. Distributed under the Open Data Commons Open Database License (ODbL) v1.0)**



**Tables**

**Table 1: EMO-5 data provider (in grey shaded the providers of gridded data).**

| Data provider acronym | Data provider name (note: name of data sets shaded in grey) | number of stations within the grid domain | precipitation | temperature | wind speed | solar radiation | vapor pressure |
|---|---|---|---|---|---|---|---|
| MARS | Wageningen Environmental Research (ALTERRA) | 5442 | x | x | x | x | x |
| EURO4M-APGD | European Reanalysis and Observations for Monitoring (EURO4M-APGD) | 4960 | x | - | - | - | - |
| MeteoConsult | MeteoConsult | 4400 | x | - | - | - | - |
| DWDSynop | Deutscher Wetterdienst | 3446 | x | x | x | - | - |
| CarpatClim | CarpatClim - Climate of the Carpathian region | 2946 | x | - | - | - | - |
| ERA Interim | European Centre for Medium-range Weather Forecasts (ECMWF) | 1402 | x | - | - | - | - |
| NMI | Norwegian Meteorological Institute | 1237 | x | x | x | - | - |
| ECA | European Climate Assessment and data set (ECA) | 1201 | x | - | - | - | - |
| EA | UK Environment Agency (EA) | 897 | x | - | - | - | - |
| AMDASynop | Deutscher Wetterdienst | 849 | x | x | x | - | - |
| ZAMG | Zentralanstalt für Meteorologie und Geodynamik | 836 | x | x | - | - | - |
| CombiPrecip | | 488 | x | - | - | - | - |
| SAIH Ebro | Confederacion hydrografica del Ebro | 386 | x | x | ? | x | - |

| ARPASIM | Agenzia Regionale per la Prevenzione e l'Ambiente dell'Emilia-Romagna | 145 | x | x | - | - | - |
|---------|-----|-----|---|---|---|---|---|
| MeteoSwiss | MeteoSchweiz | 140 | x | x | x | x | - |
| FMI | Finnish Meteorological Institute (FMI) | 110 | x | x | - | - | - |
| HMS | Hungarian Meteorological Service | 97 | x | x | - | - | - |
| IPMA | Institute for Ocean and Atmosphere, Portugal | 96 | x | x | x | x | - |
| SHMU | Slovak Hydro-Meteorological Institute | 72 | x | x | - | - | - |
| CHMI | Czech Hydro-Meteorological Institute | 69 | x | x | - | - | - |
| IMGW | Institute of Meteorology and Water Management - National Research Institute (Poland) | 50 | x | x | - | - | - |
| METIE | Met Éireann | 32 | x | x | x | - | - |
| ARSO | Slovenian Environment Agency | 21 | x | x | x | x | - |
| KHMI | Hydrometeorology Institute of Kosovo | 4 | x | x | - | - | - |


**Table 2: Overview of the five gridded meteorological data sets that form the input for EMO-5.**

| Gridded data set (reference) | Description |
|---|---|
| **ZAMG-INCA** (Haiden et al., 2011) | A near real-time high-resolution (1x1 km) multi-variable meteorological data set covering the whole of Austria. The MDCC imports from the INCA data set 6-hourly precipitation and 6-hourly mean temperature information at 836 pre-selected locations into the MDCC data collection. The data collection is done in near real-time since 2009, whereas historical data from 01.01.2003 onwards have been loaded as a bulk order. |
| **COMBI-PRECIP** (Sideris et al., 2014) | A near real-time radar / rain-gauge product for hourly precipitation covering the whole of Switzerland as well as bordering regions. 1-hourly precipitation data is imported into the MDCC data collection in near real-time at 488 virtual stations across Switzerland since 01.06.2017. |
| **EURO4M-APGD** (Isotta et al., 2014) | A daily precipitation data set from 1971 till 2008 covering the Alps and adjacent flatlands (area between 4.8 and 17.5°E as well as 43 to 49°N). The daily precipitation is imported into the MDCC data collection at 4960 virtual stations from 01.01.1971 till 31.12.2002. From 01.01.2003 the number of virtual stations imported to the MDCC data collection dropped to 4123 stations, as 837 stations over Austria were excluded due to the availability of the higher resolution INCA data covering that area. |

| **CarpatClim**<br>(Antolović et al., 2013; Spinoni et al., 2015) | Another historical daily precipitation product, covering the area between 44 and 50°E as well as 17 to 27°N (Hungary, Serbia, Romania, Ukraine, Slovakia, Poland, Czech Republic, Croatia) at a horizontal resolution of 0.1°. The data is imported into the data collection for the entire historical period from 1970 till 2010 as 2946 virtual stations. |
|---|---|
| **ERA-Interim/Land**<br>(Dee, 2011; Dee et al., 2011) | A global multi-variable reanalysis data set for all land surface areas from 1979 till October 2016 at a 0.75°x0.75° spatial resolution. The import of ERA-Interim/Land data has been limited to peripheral areas for which the coverage with in situ stations in the MDCC database is very low, such as Iceland, North Africa and the eastern part of the EFAS domain (Near East, Caucasus, Russia). For those areas, 6-hourly precipitation was imported for 1402 virtual stations. |



**Table 3: Specifications for the validation rules as applied on the EMO-5 source data**

| Variable | Monthly validation (Rule 2) | Cross validation (Rule 3) | Min/max validation (Rule 4) | | Rate of change validation (Rule 5) |
| | | | Minimum threshold | Maximum threshold | |
| --- | --- | --- | --- | --- | --- |
| **Precipitation [mm]** | yes | | 0 | • 125 (15 min)*<br>• 200 (30 min)<br>• 250 (1 h)<br>• 350 (3 h)<br>• 425 (6 h)<br>• 475 (9 h)<br>• 500 (11 h)<br>• 525 (15 h)<br>• 550 (18 h)<br>• 600 (24 h) | |
| **Mean temperature [°C]** | - | | -45 | +43 | • 10 (15, 30, 60 min)**<br>• 20 (3, 6 h)<br>• 25 (12 h)<br>• 30 (24 h) |
| **Min temperature [°C]** | - | | -50 | +35 | |
| **Max temperature [°C]** | - | | -40 | +50 | |
| **Vapor pressure [hPa]** | yes | | - | - | |
| **Solar radiation [J/m²]** | yes | | 0 | 1360 | |
| **Wind speed [m/s]** | yes | Wind speed (WS) depending on Wind direction (WD)<br>• IF WS = 0 AND WD ≠ 0 THEN WS suspect<br>• IF WS ≠ 0 AND WD = 0 THEN WS suspect | 0 | 45 | |

* the maximum threshold for precipitation depends on the aggregation interval

** the maximum change of temperature [K] depending on the observational interval (given in parenthesis)



**Table 4: Overview of the three spatial interpolation schemes that were evaluated for the purposes of EMO-5.**

| # | Interpolation scheme | Description | Parameterisation |
|---|---|---|---|
| 1. | **Inverse Distance Weighting (IDW)** | IDW is a simple and robust scheme with low computational cost. It is a purely geometric scheme based on the assumption that the closer the meteorological station is to the grid cell centre, the more related it is to its actual value. Mathematically, this is expressed by assigning weights to the surrounding stations proportional to their distance d, e.g. $1/d^2$. | • Minimum Number of stations: 4 <br> • Maximum Number of stations: 10 <br> • No restriction in search radius <br> • Weight function $\sim 1/d^2$ <br> • Initial search radius as in Shepard, 1968 |
| 2. | **Modified SPHEREMAP** | The original SPHEREMAP (Willmott et al., 1985) is the adaptation to spherical coordinates of Shepard's inverse distance weighting (Shepard, 1968), which is an extension to the IDW scheme described above. Interpolation weights decrease with increasing distance between grid centre and meteorological station, as in IDW, but the equation for the calculation of the interpolation weight depends also on the distance and number of available input stations. Furthermore, the interpolation takes the clustering of stations into account, so the weights of clustered stations were reduced in order not to overweight these data. As the original SPHEREMAP scheme would lead to a neglect of many stations in regions with a high station density we adapted the algorithm. Previously, if at least one station was found within the smallest search radius "epsilon", then this station or the mean of the stations within "epsilon" was utilised and the station outside the "epsilon" neglected. Now, the "epsilon" is set as 1/20 of the initial search radius and the distance from stations within the radius is set to "epsilon" to avoid an overweighting of the nearest station(s). With these modifications the utilisation of at least four stations per grid point is assured. The maximum number of stations used for interpolation is set to 10. | • Minimum Number of stations: 4 <br> • Maximum Number of stations: 10 <br> • No restriction in search radius <br> • Initial search radius as in Shepard, 1968 |
| 3. | **Ordinary Kriging** | Ordinary Kriging (Krige, 1966) is an advanced geostatistical method based on correlations between observations. The interpolation weights, which are created by means of the variograms, make use of observation data. Briefly, variograms sort the variance between observations by the distance between these observations were taken. Several approaches can be used to compute these variograms, such as calculations of variograms for each station and time separately, or climate zone dependent variograms, but here utilizing one global variogram for all interpolations, as is utilized at the Global Precipitation Climatology Centre / GPCC (Schamm et al., 2014). | • Minimum Number of stations: 4 <br> • Maximum Number of stations: 10 <br> • No restriction in search radius <br> • One variogram for whole domain, not season-dependent <br> • Initial search radius as in Shepard, 1968 |

**Table 5: Summary of the error measures for the three interpolation schemes based on the leave one out analysis. (Best values are in bold)**

| | ME | | | MAE | | | MSE | | |
|---|---|---|---|---|---|---|---|---|---|
| | IDW | SP | KRI | IDW | SP | KRI | IDW | SP | KRI |
| **Precipitation** | 0.89017 | **-0.01585** | -0.02312 | 2.3233 | **1.35704** | 1.4019 | 92192.22 | 12.88 | **12.304** |
| **Min temp** | 0.06418 | **0.04342** | 0.060102 | **1.6015** | 1.6164 | 1.6225 | **5.2379** | 5.592 | 5.3695 |
| **Max temp** | 0.035612 | 0.045672 | **0.00394** | **1.7648** | 1.7781 | 1.7878 | **7.2925** | 8.0828 | 7.5285 |
| **Wind speed** | 0.026899 | **0.007977** | 0.0322 | **0.9628** | 1.02 | 0.9706 | **1.9856** | 2.31 | 2.0056 |
| **Water vapour pressure** | 0.010876 | **0.002848** | 0.017395 | 0.8525 | **0.8415** | 0.85295 | 1.9404 | 1.9568 | **1.9287** |
| **Radiation** | 22.1408 | 17.2062 | **15.3328** | **2151.65** | 2283.68 | 2152.9 | **8686705** | 10074971 | 8690056 |

**Table 6: Brief product specification of EMO-5.**

| Data set description | |
|---|---|
| **Name of the data set** | EMO-5 (EMO stands for "European Meteorological Observations", whereas the 5 denotes the spatial resolution of 5 km) |
| **Short description** | EMO-5 (version 1) is a European high-resolution, (sub-)daily, multi-variable meteorological data set built on historical and real-time observations obtained by integrating data from 18,964 ground weather stations, four high-resolution regional observational grids (i.e. CombiPrecip, ZAMG - INCA, EURO4M-APGD and CarpatClim) as well as one global reanalysis (ERA-Interim/Land). EMO-5 includes at daily resolution: total precipitation, temperatures (minimum and maximum), wind speed, solar radiation and water vapour pressure. In addition, EMO-5 also makes available 6-hourly precipitation and mean temperature. The raw observations from the ground weather stations underwent a set of quality controls, before SPHEREMAP and Yamamoto interpolation methods were applied in order to estimate for each 5x5 km grid cell the variable value and its affiliated uncertainty, respectively. |
| **Created by** | Copernicus Emergency Management Service |
| **Horizontal coverage** | Europe (EFAS domain) |
| **Horizontal resolution** | 5 km |

| | |
|---|---|
| **Spatial gaps** | Only land areas are covered by this data set |
| **Temporal coverage** | 1990-01-01 till 2019-12-31 |
| **Temporal resolution** | daily and 6-hourly |
| **Temporal gaps** | No gaps |
| **Number of available variables** | 7 |
| **Variables available at daily resolution** | total precipitation, temperatures (minimum and maximum), wind speed, solar radiation and water vapour pressure |
| **Variables available at 6-hourly resolution** | total precipitation and mean temperatures |
| **Units** | precipitation [mm], temperature [°C], vapor pressure [hPa], solar radiation [J/m²], wind speed [m/s] |
| **Projection** | Lambert Azimuthal Equal-Area (5km) |
| **Data type** | 5*5 km grids |
| **Available version** | version 1 |
| **DOI of dataset** | https://doi.org/10.2905/0BD84BE4-CEC8-4180-97A6-8B3ADAAC4D26 |
| **PID of dataset** | http://data.europa.eu/89h/0bd84be4-cec8-4180-97a6-8b3adaac4d26 |

**Table 7: Overview of EMO-5, seNorge2 and seNorge2018 data sets based on the common period 1990-2015 [mm]. The order of calculation is first to calculate the parameters (mean, maximum) for the grids, followed by the mean over the resulting grid.**

| | EMO 5 | | | | | SeNorge2 | | | | | SeNorge 2018 | | | | |
|---|---|---|---|---|---|---|---|---|---|---|---|---|---|---|---|
| | **Ann** | MAM | JJA | SON | DJF | **Ann** | MAM | JJA | SON | DJF | **Ann** | MAM | JJA | SON | DJF |
| **mean** | 968 | 183 | 248 | 279 | 255 | 948 | 179 | 247 | 274 | 247 | 1163 | 227 | 266 | 328 | 340 |
| **max** | 1371 | 346 | 423 | 496 | 467 | 1291 | 317 | 406 | 471 | 433 | 1601 | 424 | 446 | 573 | 603 |