# Peer review of "EMO-5: A high-resolution multi-variable gridded meteorological data set for Europe"

_Earth System Science Data, 2021_

## Referee Comment (RC1)

**Review of 'EMO-5: A high-resolution multi-variable gridded meteorological data set for Europe' by Vera Thiemig et al.**

The authors intend to introduce and describe a gridded high resolution dataset over Europe produced from in-situ sub-daily meteorological observations supplemented by information from reanalyses and various other products. The dataset is used by CEMS in support of operational real-time emergency management and in principle could have many other applications. The dataset description paper is therefore, in principal, a useful contribution to the literature.

Aspects around data collection, quality assurance and gridding / interpolation are undoubtedly valuable. However, in almost all cases they fall consistently short of state-of-the-art practices undertaken elsewhere and, therefore, it is hard to justify claims that this is a uniquely valuable dataset. There is considerable room for future improvement in all aspects of the analysis.

That said, there is also value in version pointing at this point and thus publishing the current approach as an initial basis upon which further improvements can subsequently be made. To do so will require substantive revisions to the present manuscript to:
  I.   Better document key aspects
  II.  Apply caveats and draw comparisons to state-of-the-art approaches as relevant
  III. Describe far more holistically how to access the data, its available formats, file structures, version control etc.
  IV.  Incorporate a substantive discussion of future requirements and present limitations

The journal is undoubtedly the appropriate place to publish this work. But the authors would need to redraft comprehensively addressing all the enumerated major points that follow for me to be able to recommend eventual publication. Given the volume of major points which will require substantive redrafting I will not make additional minor comments at this juncture.

**Major points**
  1. The introduction is too short. Greater effort is required on scene setting including a discussion of the role and limitations of both reanalysis and satellite observations to provide the reader with necessary context to then properly interpret the paper.

     Indeed, there is in general a lack of sufficient acknowledgement of prior and ongoing valuable work in this area across the submitted manuscript as a whole to provide the reader with necessary context to properly and fairly evaluate the value of your work. For example ECA&D and the associated E-OBS gridded product served via the C3S CDS should be noted much more prominently and far earlier and calls into substantial question your assertion on lines 39-41. I would be hard pressed to defend the statement you make there given the availability of the well documented and highly utilised E-OBS product. Similarly, the efforts under C3S to collate station data as documented in e.g. Noone et al., 2021 warrants more attention than is currently given. Failing to acknowledge these – Copernicus funded – efforts just gives an impression of dysfunctionality across Copernicus program activities as well as –

for those with the knowledge – giving the distinct impression of over-selling what you have produced and doing so in glorious isolation. This is disappointing as a knowledgeable reviewer and equally does not bode well for realising programmatic synergies across Copernicus services in the new Copernicus budgetary cycle activities.

The paragraph starting at line 55 should come earlier and be expanded. Then in numerous places comparisons to existing products / approaches should be made. For example, the QC method in 3.1 should make reference to QC procedures of E-OBS, GHCND and HadISD at a minimum.

2. The reason why the analysis is limited to 1970 needs to be made far more explicit. There are many data that extend back further than 1970 and the choice of 1970 is arbitrary with no clear rationale given to justify the choice. From an application perspective there would be considerable value to extending the analysis to earlier periods to the extent possible.

   Given that the product is reputedly available from 1970 onwards it is problematic that Figures 1-3 only show availability post-1990. This gives the impression that you may be wishing to hide / obscure availability of data in the first 20 years of the series. The figures should show 1970 onwards or the product should be cut at 1990 and published as such. Later at the end of section 3 you suggest this is the case. Please clarify whether the product extends 1970 to date or 1990 to date and ensure consistently stating so throughout the paper.

3. The methodology in almost all aspects needs to be strengthened and much more explicit. The method must be outlined to the extent that the analysis could be to first order independently reproduced based upon the description. This is necessary and a central tenet of the scientific method itself. Please revise the method to be considerably more comprehensive such that based upon the method description an independent researcher may reasonably be able to recreate your approach.

4. The validation activity would be stronger if it could be shown in more detail for all variables and would, undoubtedly, serve to increase user uptake. However, the only real validation in the strict scientific sense is actually the leave-one-out analysis which is not even contained within the validation section at all. What is presently classed as being validation is actually characterisation of the product. Validation requires a value that is of known quality to compare against and none of the three sub-sections in the section purportedly to do with validation actually undertake such a comparison. That section should be retitled to Characterisation accordingly. Then the leave-one-out validation should be considerably expanded, shown for numerous parameters, and aggregated across well-sampled and sparsely-sampled regions separately.

5. In section 2 the input data is outlined to a very perfunctory level – greater detail here would help including a breakdown by source, data policy etc and links to sources where available publicly by elevating Table A1 to the main text and

augmenting with additional information. Also, the maps in Figures 1 to 3 conflate station and gridded input in ways that are really inaccessible. Maps of just primary station source by time and / or variable would be more accessible and avoid proverbially mixing apples and oranges together.

Worse, Section 2 is also completely silent on sharing of these station data onwards to e.g. the World Data Centre for Meteorology at NOAA NCEI or the C3S in-situ database effort or ECA&D and thus E-OBS. This limits the potential utility of this data collection activity if you are not actively trying to share data where you can with the activities being undertaken to improve access globally to meteorological holdings, many of which are funded by Copernicus. Where is the joined up thinking? How does this serve Copernicus if you are not actively sharing across Copernicus Services?

6. The quality control is a little limited, in particular in not including any form of neighbour buddy checking as is state of the art in e.g. GHCND or HadISD. Either that or, between the text and the table, the presence of buddy checks is obscured. You should at a minimum be explicit that the applied checks consist of a minimum set of record-based, logic and spike-based checks and do not include a number of other checks including repeat strings, distributional checks, frequent value checks and neighbour-based checks as is the case in state-of-the-art QC procedures such as GHCND or HadISD. You should aim to incorporate a broader array of such checks in future.

   Furthermore, I would expect the discussion of QC in section 3.1 to include some summary of the frequency with which different values are flagged and some consideration of heterogeneity of flagging of values across sources and regionally. Does the frequency of QC test failure look reasonable? Does it raise any flags for particular sources etc. etc. are all questions I would expect this paper to address if it is to build user confidence in applicability of the resulting product and yet it is presently silent on these issues.

7. The interpolation scheme discussion in 3.2 requires substantive additional detail. No discussion is forthcoming around why the different schemes have different skill per parameter. This must, intrinsically, be to do with the spatio-temporal correlation structures of the different parameters and how effectively the three schemes can handle these. In particular the parameters that vary more smoothly in space and time clearly are better suited to IDW whereas the much smaller scale correlation structure precipitation is better suited to SPHEREMAP. The lack of geophysical interpretation of the findings does not build the necessary user confidence that you understand what underlies your methods.

   The analysis should also remark upon the similarity in skill measures for most variables and diagnostics as this implies that interpolation choice isn't a first order effect presumably? Furthermore, is there any gradation in skill between well sampled and sparsely sampled regions? One would presume so but your present aggregation just lumps all cases together. It would be considerably more informative

to split this analysis and repeat for well-sampled and sparsely-sampled regions. I would expect no impact of choice in well-sampled regions where interpolation choice makes little difference but much larger impacts in sparsely sampled regions. Can you repeat that analysis but splitting out by regions of greater and lesser station density? What does that tell you?

8.  Given that ERA-Interim/Land values may be biased in terms of variance and mean state isn't the time-varying use of this product to infill in data sparse regions potentially problematic for long-term product homogeneity? Also, ERA5-Land has replaced ERA-Interim -land having higher resolution and improved quality. Should the algorithm not use this product instead? In particular because ERA-Interim land is no longer produced in CDAS mode introducing potentially a major discontinuity into your product upon cessation of ERA-Interim land production?

9.  The assumptions around 06-18 and 18-06 for Tn and Tx will miss many true Tx and Tn values particularly in the winter half season when daily maxima or minima often occur outside these windows. Particularly so in the higher latitudes of the domain. They also risk introducing a discontinuity with the mean temperatures calculated over the full 24 hours.  At a minimum caveats need to be added but, ideally, you would calculate Tx, Tn and Tm consistently for full 24 hour periods to enable cases where Tx and / or Tn fall outside of nominal day / night.

10. More details are required upon the land-sea mask used and how it is aggregated to the 5km grid. Are stations omitted based upon whether land or ocean at the native e.g. 5 minute resolution of the mask but then the gridded product produced for all 5km gridcells that contain any land? The method is not reproducible or understandable absent of such details.

11. Section 3.3.4 is a little basic compared to state-of-the-art spline fitting techniques as used e.g. in HadEX or CRUTS. Why has such a relatively speaking simple approach been used as is described in 3.3.4 and how does it compare to spline fitting or other techniques?

12. The mean temperature discussion in 3.3.5 would benefit from discussing various pieces of literature about how to create mean temperatures and the difference between the mean of a 24 hour period, the mean of max / min and various other alternative methods. There have been several papers on the different ways to calculate means and the random and systematic effects they can have. For the purpose of your dataset the key issue is whether the different ways you have done so may impart systematic or random effects. This can be ascertained from a representative (climatologically) set of well-sampled stations being deliberately degraded to calculate the mean from the 24(+) instantaneous values and then various approximations. This can then at least bound your uncertainties.

13. I would expect considerably more in the data availability section. It is grossly insufficient to just point to a website. You must describe things such as the data format(s) (ideally there should be at least two to cater to a range of users), any

software tools available, version control, version retention policies, whether files are available as spatial or temporal aggregations, whether the files contain uncertainty information, whether they are univariate or multivariate. Otherwise no user is going to go to that site on the off chance. You are not really helping yourselves to advertise the availability if all you do is point to a URL. Users need to know what they are being offered and how well it is ultimately managed.

14. seNorge2 and seNorge2018 are as I understand it consecutive versions of a single dataset with 2018 replacing 2. You should use just the 2018 version as the other version has been deprecated. Having also reviewed that paper and in particular considering their discussion it would furthermore not be appropriate to treat that product as truth given the substantial caveats made around interpolation in data sparse areas of complex topography. Therefore the seNorge2018 validation must be noted to be conditional on the verity of that product which cannot be assured a priori. You cannot treat either seNorge version as 'truth' which significantly inhibits their value as a means of validation. There are many other national gridded products and comparison to a range of such products e.g. the UK analysis from the Met Office may be instructive and reduce the dependency upon a single product which may itself contain substantive systematic biases.

15. I cannot make head or tail of Figure 11 from the caption and the individual images. The caption needs expanding and the panels given better titles. I cannot understand why the panels in each case vary in the manner they do and it is non-intuitive to me and therefore will be so to your readers. A figure extending over several pages is also barely legible. You would probably be better to concentrate upon 1 or 2 events that could be given more space in the main text and place the remaining cases in supplemental materials.

16. Lacking in entirety is any meaningful discussion section which may highlight caveats, future work avenues, future priorities, synergies with other Copernicus and global activities (such as WMO) that may yield improvements etc. etc. I would expect a substantive discussion section which covered such points.

---

## Referee Comment (RC2)

**Review of the paper** *"EMO-5: A high-resolution multi-variable gridded meteorological data set for Europe"* **by Vera Thiemig , Goncalo N. Gomes, Jon O. Skøien, Markus Ziese, Armin Rauthe-Schöch, Elke Rustemeier, Kira Rehfeldt, Jakub P. Walawender, Christine Kolbe, Damien Pichon, Christoph Schweim and Peter Salamon**

The paper describes EMO-5, a collection of gridded datasets for seven daily aggregated variables. The dataset covers Europe. The time period ranges from 1990 to 2019 and the grid spacing is 5 km. I understood that originally the aim of the developers was to provide the atmospheric forcings to a hydrological rainfall-runoff model, then EMO-5 was also distributed as a stand-alone product and made available to the general public. For this reason, the authors consider precipitation as the most important of the seven EMO-5 variables.

The input data is a blending of point observations, grid points from both observational gridded datasets and reanalyses. This implementation choice is one of the most interesting aspects of the work and -at the same time- one of its critical points.

Three different statistical interpolation schemes have been considered for EMO-5 production and SPHEREMAP was chosen as the best one. At the same time, the comparison of the three schemes allows the authors to present an evaluation of the products based on leave-one out cross-validation. The authors compare EMO-5 against seNorge over the Norwegian mainland. Then, 15 extreme events have been presented as case studies and EMO-5 has been evaluated qualitatively over them.

I believe that the topic of the paper is of extreme interest. Data fusion experiments, such as the one presented here, meet the demand of many users of weather data, which is to bridge that last mile that allows them to use the best input in a given application. In fact, the best input is often a combination of data sources. The weak point of the paper is that in its present form many of the claims the authors make in their conclusions are not sufficiently reflected in the results presented.

In conclusion, I think the manuscript can be published once the following comments are resolved.

Major Comments:

- *What is a "best" estimate for you?* In the paper, the concept of best estimate (or best guess) is often mentioned. The authors should explicitly state, in the Introduction, in what sense a predicted value is "the best" for them. This "declaration of intent" can be useful for some users to decide if your dataset is suited for their applications. From the text, It seems that in data dense regions the best estimate is an average of the observations, while in data sparse regions the accuracy and precision of the representation will be the same as for ERA-Interim. Note that other definitions are possible, for instance it might be that the best prediction is the one that allowed the authors to get the most realistic output out of a hydrological model.

  As a further remark that shows the importance of understanding what is a best estimate for you, I may discuss the reasons that led you to discard ordinary Kriging as the optimal interpolation method. Ordinary Kriging implements a spatially

consistent low-pass filter over the entire domain. The properties of the filter vary in time, because they are defined by the time-varying semi-variograms. However, for a given day, ordinary Kriging provides results that are much more comparable between different regions across Europe because the "smoothing" makes them more comparable. On the other hand, the smoothing filters out the local scales that one can represent in data-dense regions, which is where the 4000 observations used for evaluation most likely are located. The spatial coherence provided by Kriging is not among the properties of the optimal "best estimate" that is interesting for the authors, then ordinary Kriging is discarded.

- *Validation approach.* The validation focuses almost exclusively on precipitation. The other variables are not considered, except for Table 4. A further weakness of the current validation approach (Table 4) is that the whole domain is considered simultaneously and only aggregated results are presented. However, as written in the previous comment, the predicted values do have very different meanings across the domain. A distinction between dense- and sparse- data regions is required to have an idea of the actual quality of the fields. The validation against seNorge is difficult to interpret (and representative of precipitation only): in some regions precipitation is smaller than seNorge, in others it is higher. Then, the conclusion is that EMO-5 is in good agreement with seNorge2, which seems not to be the case for some climatological indices you present (i.e. CDD, CWD in Fig. 10). Finally, the evaluation over the case studies is only qualitative: for instance, does the spatial distribution of the observations have an influence on the way extremes are represented?

I may suggest a different approach that could solve most of the problems I see with the current approach. I have the impression that the aim of the validation is to show that EMO-5 is fit-for-purpose for hydrological simulations. If this is the aim, I think this should be stated explicitly. The authors should present their results on precipitation separately for observational dense and sparse regions. Furthermore, the authors should include a section on "indirect validation" of the meteorological fields, where the output of a hydrological model that is using EMO-5 fields as forcings is evaluated. In this way, all the seven variables will be evaluated simultaneously on their potential in providing useful results for the application they were designed to serve.

- *Interpolation uncertainty.* I have never mentioned interpolation uncertainty until now. I think that the authors do not provide enough evidence that the uncertainty fields they provide are actually characterizing the actual uncertainty in a reliable way. The only results they show is Figure 4, linked to a short discussion in Sec. 5.1. A proper validation of the uncertainty field would require the presentation of much more results (see e.g. Wilks (2019), Chapter 9). In my opinion, the status of the EMO-5 uncertainty products are "still under development" and they can be the topic of future works. Furthermore, if I understand well, the uncertainty is provided for all of the 7 variables but in the paper it is presented only for precipitation. In its present form, I suggest the paper should not mention the interpolation uncertainty as a product available to the users.

Minor comments:

- *Verification scores used in Sec. 3.2.* The verification scores considered in the cross-validation exercises are not well suited for the validation of precipitation, especially over very large areas. I would have used something like the Nash–Sutcliffe model efficiency coefficient (NSE) or the Equitable threat score (ETS) for some significant thresholds (e.g. 1 mm/day, 10 mm/day,....).

- *Interpolation methods.* It is not clear why SPHEREMAP is used for all variables. Since IDW is better for some variables, then why not use it for those variables?

- *Quality control.* The division between "suspect" and "good" data is not used at all, since EMO-5 uses both data. Then, why are the QCs flagging some data as suspicious?

---

## Author Response (AR1)

**Reviewer2**

**Review of the paper** *"EMO-5: A high-resolution multi-variable gridded meteorological data set for Europe"* **by Vera Thiemig , Goncalo N. Gomes, Jon O. Skøien, Markus Ziese, Armin Rauthe-Schöch, Elke Rustemeier, Kira Rehfeldt, Jakub P. Walawender, Christine Kolbe, Damien Pichon, Christoph Schweim and Peter Salamon**

The paper describes EMO-5, a collection of gridded datasets for seven daily aggregated variables. The dataset covers Europe. The time period ranges from 1990 to 2019 and the grid spacing is 5 km. I understood that originally the aim of the developers was to provide the atmospheric forcings to a hydrological rainfall-runoff model, then EMO-5 was also distributed as a stand-alone product and made available to the general public. For this reason, the authors consider precipitation as the most important of the seven EMO-5 variables. The input data is a blending of point observations, grid points from both observational gridded datasets and reanalyses. This implementation choice is one of the most interesting aspects of the work and -at the same time- one of its critical points.

Three different statistical interpolation schemes have been considered for EMO-5 production and SPHEREMAP was chosen as the best one. At the same time, the comparison of the three schemes allows the authors to present an evaluation of the products based on leave-one out cross-validation. The authors compare EMO-5 against seNorge over the Norwegian mainland. Then, 15 extreme events have been presented as case studies and EMO-5 has been evaluated qualitatively over them.

I believe that the topic of the paper is of extreme interest. Data fusion experiments, such as the one presented here, meet the demand of many users of weather data, which is to bridge that last mile that allows them to use the best input in a given application. In fact, the best input is often a combination of data sources. The weak point of the paper is that in its present form many of the claims the authors make in their conclusions are not sufficiently reflected in the results presented.

In conclusion, I think the manuscript can be published once the following comments are resolved.

Dear Reviewer,

we have appreciated your comments and suggestions made, and have addressed each one of them. Please find below a detailed reply to each point. Further, we have submitted a new version of the manuscript using track-changes to show transparently were we have made changes.

Best wishes,

Vera Thiemig on behalf of all the authors

**Major Comments:**

1. *What is a "best" estimate for you?* **In the paper, the concept of best estimate (or best guess) is often mentioned. The authors should explicitly state, in the Introduction, in what sense a predicted value is "the best" for them. This "declaration of intent" can be useful for some users to decide if your dataset is suited for their applications. From the text, It seems that in data dense regions the best estimate is an average of the observations, while in data sparse regions the accuracy and precision of the representation will be the same as for ERA-Interim. Note that other definitions are possible, for instance it might be that the best prediction is the one that allowed the authors to get the most realistic output out of a hydrological model.**

   **As a further remark that shows the importance of understanding what is a best estimate for you, I may discuss the reasons that led you to discard ordinary Kriging as the optimal interpolation method. Ordinary Kriging implements a spatially consistent low-pass filter over the entire domain. The properties of the filter vary in time, because they are defined by the time-varying semi-variograms. However, for a given day, ordinary Kriging provides results that are much more comparable between different regions across Europe because the "smoothing" makes them more comparable. On the other hand, the smoothing filters out the local scales that one can represent in data-dense regions, which is where the 4000 observations used for evaluation most likely are located. The spatial coherence provided by Kriging is not among the properties of the optimal "best estimate" that is interesting for the authors, then ordinary Kriging is discarded.**

   Reply to major point 1:

   We have re-written the introduction to put transparency on our "declaration of intent" as the reviewer calls it. EMO-5 is not only suitable or intended for the use in hydrological modelling. It is already now used within the European Drought Observatory, the European Forest Fire Information System, and several JRC in-house applications focussing on agricultural modelling. We have elevated this information in the introduction

   In Section 3.2, we state how we are going to select our interpolation scheme: based on the reliability, specifically regarding uncertainty and computational cost. The uncertainty is evaluated through three different measures of errors: Mean Error (ME), Mean Absolute Error (MAE) and Mean Squared Error (MSE), each focusing on different aspects of uncertainties. The computational costs increases on the one hand with the number and complexity of interpolation algorithms that are implemented and need to be maintained, and on the other hand with the computational time.

As EMO-5 is an operational near real-time data set which requires a robust computational framework that is fairly easy maintainable, we have decided to select only one interpolation scheme that works for all variables, rather than choosing for each variable the optimal interpolation scheme, which would easily end up in a multitude of algorithms that are complex to maintain in a fully operational near real-time environment.

The 4000 stations that were used for the leave-one-out validation were randomly chosen over various clusters. Clusters were used to make sure that the randomly chosen stations do not concentrate in areas with a high station density, but rather distribute across the entire domain including low station density areas.

As precipitation is of highest importance in hydrologic modelling and many other environmental models, a higher weight was given to the interpolation performance of precipitation compared to the other parameters. This might post the question why we have included the other variables at all in this analysis. The answer is that we wanted to be sure that the chosen interpolation scheme is a good solution for all the variables, even if for some variables another interpolation scheme would outperform the one working the "best" for precipitation.

Ordinary Kriging has the highest computational cost, and is less robust in an operational framework, as it would require an automatic fitting of the time-varying semi-variograms which had been found problematic (see p. 8 of Ntegeka et al., 2013: DOI: 10.2788/51262). Hence, the decision for the interpolation scheme was driven by the application side and not to characterize interpolation schemes.

Changes to the manuscript:
a) We have re-written the introduction and have emphasised the current and potential applications for EMO-5, to show that it is no exclusively for hydrological modelling. Please see the entire introduction, but here the most relevant section:
   "The (timely) availability of meteorological data and their quality determine in many cases the capacity of environmental modelling [...] such as e.g. hydrological or agricultural models, as well as environmental and risk indicators."
   [...]
   "The CEMS MDCC runs around the clock and produces daily near real-time meteorological grids which are used in the operational running of not only EFAS, but also by two other major CEMS services, namely the European Forest Fire Information System (EFFIS; https://effis.jrc.ec.europa.eu/; San-Miguel, J. et al., 2019) and the European Drought Observatory (EDO; https://edo.jrc.ec.europa.eu/; Spinoni et al., 2016; Cammalleri, C. et al., 2020)."

b) We have elevated the information on the choice of the interpolation scheme (see section 3.2), here an excerpt:
   "The quality of each of the interpolation schemes was derived through a leave-one-out cross-validation. This means that for each iteration of the interpolated field, one station was left out and then later on compared with its interpolated value. This was done for around 4000 randomly chosen stations evenly distributed over high and low density station areas, and those pairs of interpolated and real observations were used to compute the uncertainty

estimates. A similar approach was applied by Hofstra et al. (2008) for the E-CAD data set."
[...]
"Due to stability concerns with the automatic variogram fitting in Kriging (Ntegeka et al., 2013), which is in particular a concern in an operational environment and the fact that none of the tested schemes outperformed the others, SPHEREMAP was chosen as the interpolation scheme to generate the grids of EMO-5, as it shows the best performance for the critical parameter precipitation."

The following reference was added:
Ntegeka, V., Salamon, P., Gomes, G., Sint, H., Lorini, V., Zambrano-Bigiarini, M., and Thielen, J.:EFAS-Meteo: A European daily high-resolution gridded meteorological data set for 1990 – 2011, JRC Technical Reports, EUR26408, DOI: 10.2788/51262, 2013.

2. ***Validation approach.*** **The validation focuses almost exclusively on precipitation. The other variables are not considered, except for Table 4. A further weakness of the current validation approach (Table 4) is that the whole domain is considered simultaneously and only aggregated results are presented. However, as written in the previous comment, the predicted values do have very different meanings across the domain. A distinction between dense- and sparse- data regions is required to have an idea of the actual quality of the fields.**

**The validation against seNorge is difficult to interpret (and representative of precipitation only): in some regions precipitation is smaller than seNorge, in others it is higher. Then, the conclusion is that EMO-5 is in good agreement with seNorge2, which seems not to be the case for some climatological indices you present (i.e. CDD, CWD in Fig. 10). Finally, the evaluation over the case studies is only qualitative: for instance, does the spatial distribution of the observations have an influence on the way extremes are represented?**

**I may suggest a different approach that could solve most of the problems I see with the current approach. I have the impression that the aim of the validation is to show that EMO-5 is fit-for-purpose for hydrological simulations. If this is the aim, I think this should be stated explicitly. The authors should present their results on precipitation separately for observational dense and sparse regions. Furthermore, the authors should include a section on "indirect validation" of the meteorological fields, where the output of a hydrological model that is using EMO-5 fields as forcings is evaluated. In this way, all the seven variables will be evaluated simultaneously on their potential in providing useful results for the application they were designed to serve.**

Reply to major point 2:

We appreciate the reviewers comment and suggestion with regards to the validation approach used and would like to explain our choice for validation approach in more detail. Even though that EMO-5 originated about 16 years ago from the need to drive the European Flood Awareness System, it is already now used by the European Drought Observation and the European Forest Fire Information System, and it's currently tested in-house (JRC) for several other agricultural applications. Hence, its application has already propagated beyond the hydrological modelling frame and therefore we chose a more holistic validation approach, looking into the data set from various angles (comparison against a high-res. data set, extreme precipitation events, interpolation uncertainties), but with the focus on precipitation, as a) this is the main driver for a large range of environmental models, hydrological models being one of them and b) EMO-5 minimum and maximum temperature have been already investigated by Lavaysse et al. (2018), they referred to EMO-5 as LisFlood in their paper for the simple reason that we had never branded our meteorological data set beforehand, as we have not published it up to now, and so they used the name of our hydrological model to name it. Hence, we hopefully can agree that with temperature already being covered, that focussing on precipitation is acceptable, also because the current publication is already fairly lengthy.

The proposed comparison of the interpolation schemes considering the station density is an additional study and beyond the scope of this paper. A similar study was presented by GPCC in a paper from 2012 at the EGU General Assembly only for precipitation. Briefly, the interpolation uncertainty decreases with increasing station availability. The interpolation schemes didn't depict the same sensitivity, a scheme with the 'lowest' uncertainty at low station density had the 'highest' uncertainty at high station density and vice versa.

Changes to the manuscript:

a) We have re-written the introduction to shed more light on how CEMS Meteorological Data Collection Centre started and what EMO-5 is (an operational data set and not a climatological data set), our reasons for publishing it, the objective and limitations of the current study, including an encouragement of further/ external review studies. Please see new Introduction section, here just one part:
"In this paper, the evaluation of the resulting grid quality is focused mainly on the gridded precipitation data, as a) precipitation is the most crucial driver for hydrological modelling (our main focus) and b) the minimum and maximum temperature of EMO-5 have been already investigated by Lavaysse et al. (2018) (EMO-5 was referred to as LisFlood in their study). However, we invite the scientific community to expand the evaluation exercise beyond EMO-5 gridded precipitation to other variables and other validation approaches including through various environmental applications."

b) We have referenced to the already existing study that evaluates EMO-5 minimum and maximum temperature.
Lavaysse, C., Cammalleri, C., Dosio, A., van der Schrier, G., Toreti, A., and Vogt, J.: Towards a monitoring system of temperature extremes in Europe, Nat. Hazards Earth Syst. Sci., 18, 91–104, https://doi.org/10.5194/nhess-18-91-2018, 2018.

3.  ***Interpolation uncertainty.* I have never mentioned interpolation uncertainty until now. I think that the authors do not provide enough evidence that the uncertainty fields they provide are actually characterizing the actual uncertainty in a reliable way. The only results they show is Figure 4, linked to a short discussion in Sec. 5.1. A proper validation of the uncertainty field would require the presentation of much more results (see e.g. Wilks (2019), Chapter 9). In my opinion, the status of the EMO-5 uncertainty products are "still under development" and they can be the topic of future works. Furthermore, if I understand well, the uncertainty is provided for all of the 7 variables but in the paper it is presented only for precipitation. In its present form, I suggest the paper should not mention the interpolation uncertainty as a product available to the users.**

    Reply to major point 3:

    The interpolation uncertainty should be a measure of the accuracy of the estimated values. It could also be called interpolation inaccuracy, but to our knowledge the term uncertainty is more frequently used.
    A leave-one-out approach was used when selecting the best performing interpolation scheme for all variables and several scores, namely the mean error (ME), mean absolute error (MAE) and mean squared error (MSE). Such a leave-one-out approach has too much computational effort to be used in the operational grid creation, even if it would sheds good light on the uncertainty. Therefore, and as described in section 3.2, an alternative technique was searched to provide also information about the reliability of the grids. As it could be expected that the accuracy of an estimated value decreases with increasing variance of the input data, the standard deviation of the used data should give a rough estimation of the reliability. This idea was refined by Yamamoto (2000), as the station data are weighted with their interpolation weights. A weakness of this technique is, that the calculated uncertainty could be too small in sparsely sampled regions. Nevertheless those uncertainty information provide a good estimation of areas/regions in the interpolated maps where additional station data would add more value.
    We presented uncertainty maps only for precipitation for consistency as all data examples are about precipitation.

**Minor comments:**
1.  **Verification scores used in Sec. 3.2. The verification scores considered in the cross-validation exercises are not well suited for the validation of precipitation, especially over very large areas. I would have used something like the Nash–Sutcliffe model efficiency coefficient (NSE) or the Equitable threat score (ETS) for some significant thresholds (e.g. 1 mm/day, 10 mm/day,....).**

    Reply to minor comment #1:

Thank you very much for your suggestions. The Nash-Sutcliffe model efficiency coefficient (NSE) was previously known to us more in the sense of discharge than in the sense of precipitation, but it is very interesting to think about using it in our context. From a pure calculation point of view, the NSE is quite similar to the Mean Squared Error (MSE) we use. The main difference is a kind of "standardisation" in the case of the NSE, since it is additionally divided by the variance of the observations. This naturally gives a very good estimate of whether the differences between model and observation are larger or smaller than the variance. However, in terms of cross-validation, where we compare different interpolation methods, our main focus is not on the variance. The assessment of which method works better and which less well should provide the same results for NSE as for the MSE because of the similar calculation of the NSE. Together with Mean Error (ME) and Mean Absolute Error (MAE), this gives a very good overall picture. The MSE is also more comparable with the other scores, as it has the same "units" as ME and MAE.

In our opinion, the Equitable Threat Score (ETS) is not suitable for cross-validation. This is because it is based on the probability table and, as you described, is only calculated for different threshold values. In a cross-validation it would need a single score that can be described by one value and not several depending on threshold values. You are of course right that the ETS is very suitable for assessing precipitation. Nevertheless, in the seNorge comparisons, we focused on the extreme value indices ETCCDI instead of the ETS, since the extremes are of particular importance for floods.

2. **Interpolation methods. It is not clear why SPHEREMAP is used for all variables. Since IDW is better for some variables, then why not use it for those variables?**

Reply to minor comment #2:

This is a very valid comment. The decision to use only one interpolation scheme for all variables was driven by the application side and not to characterize interpolation schemes. As EMO-5 is an operational data set with many variables that is created in near real-time we needed an optimal interpolation scheme in terms of computational effort, computational time and lowest uncertainty that works for all variables. As precipitation is of highest importance in hydrologic modelling, a higher weight was given to the performance in precipitation than the other parameters. For precipitation, SPHEREMAP outperformed the other tested interpolation schemes, and was therefore used as interpolation scheme for the entire data set.

Changes to the manuscript:
"In favour of obtaining an operational framework that is maintainable, the decision was taken to choose only one interpolation scheme for all variables, given that the resulting grid quality of all variables allows this. The analysis of the error measures shows that none of the tested interpolation schemes outperforms the others in a consistent manner. However, working within an operational environment, the robustness of the system is crucial, and from that point of view, the automatic variogram fitting in Kriging is a stability concern that was also observed by Ntegeka et al. (2013), and hence was excluded for EMO-5. As SHPEREMAP shows the best performance for the critical

parameter precipitation, it was ultimately chosen as the interpolation scheme to generate the grids of EMO-5."

3. **Quality control**. The division between "**suspect**" and "good" data is not used at all, since EMO-5 uses both data. Then, why are the QCs flagging some data as suspicious?

Reply to minor comment #3:

We fully agree that it this seems confusing and we will try our best to shed light on why that is. A value is flagged as suspect if it could be wrong, but might not be. There are few and well defined cases when a data value is flagged as "suspect" as explained in the last paragraph of Section 3.1:

A data value is flagged as […] "suspect" if the time stamp has been corrected (Rule 1). It is also flagged as "suspect" if it fails the validation against the monthly statistics (rule 2) or the cross-validation (rule 3).

The one on the time-stamp correction is probably the easiest one to agree with, while the other two it is harder to get ones head around, but let me explain. The cross validation is only done for wind speed and wind direction. If a station reports 0 wind speed, but not 0 on the wind direction, then both are flagged as suspect, and vice versa if the wind direction is 0, but the wind speed not, then that is also flagged as suspect. While values are flagged suspect if they are out of the bounds of the monthly validation, but each values within that time period passes the min/max validation, and hence it is "suspect" that the monthly value is out of the station typical boundaries, whereas all the individual values are within the min/max thresholds.

In-house experiments have shown that the quality of resulting grids is higher when using the "suspect" values, and lower when they are removed.

**Reply to reviewer 1 of 'EMO-5: A high-resolution multi-variable gridded meteorological data set for Europe' by Vera Thiemig et al.**

Dear Reviewer,

we are please to share with you that we have gone through all your comments and addressed them. Please find below our reply to each major point you raised.

**Major points**
**1. The introduction is too short. Greater effort is required on scene setting including a discussion of the role and limitations of both reanalysis and satellite observations to provide the reader with necessary context to then properly interpret the paper. Indeed, there is in general a lack of sufficient acknowledgement of prior and ongoing valuable work in this area across the submitted manuscript as a whole to provide the reader with necessary context to properly and fairly evaluate the value of your work. For example, ECA&D and the associated E-OBS gridded product served via the C3S CDS should be noted much more prominently and far earlier and calls into substantial question your assertion on lines 39-41. I would be hard pressed to defend the statement you make there given the availability of the well documented and highly utilised E-OBS product. Similarly, the efforts under C3S to collate station data as documented in e.g. Noone et al., 2021 warrants more attention than is currently given. Failing to acknowledge these – Copernicus funded – efforts just gives an impression of dysfunctionality across Copernicus program activities as well as – for those with the knowledge – giving the distinct impression of over-selling what you have produced and doing so in glorious isolation. This is disappointing as a knowledgeable reviewer and equally does not bode well for realising programmatic synergies across Copernicus services in the new Copernicus budgetary cycle activities.**
**The paragraph starting at line 55 should come earlier and be expanded. Then in numerous places comparisons to existing products / approaches should be made.**
**For example, the QC method in 3.1 should make reference to QC procedures of EOBS, GHCND and HadISD at a minimum.**

Reply to point 1:

Despite that we do not think that the review of a scientific paper is a discussion platform for Copernicus internal management matters, we recognise that this is truly a concern for the reviewer. Therefore we would like to reassure him (her) that there is good collaboration ongoing between the various Copernicus core services and that we seek synergies and collaborations wherever possible. In general, the dialogue between CEMS and C3S has always existed, and efforts on mutual-beneficial solutions have been a focus. In fact, this very manuscript was shared with the entrusted entity of C3S some months ago, and was

only submitted after addressing all their comments. Hence, there is transparency and collaboration established between C3S and CEMS, including in this matter.

It might on first glance appear like a service duplication within Copernicus, however the meteorological data collection services of C3S and CEMS serve two very different needs, which principal their requirements and result into two very different products, each with their rightfulness to exist. On the one hand, CEMS needs an operational real-time dataset that contains as much valid information as available at near real-time to compute the initial state of the system, and in addition some decades of historical data created with the same method as the real-time data to allow calibration and validation of the operational system and consistency between the calibration of the model and the operational running of the forecast. On the other hand, C3S serves the community with a climatological dataset, and hence their data products need to be of sufficient length, consistency, and continuity to determine climate variability and climate change. It appears that this distinction was not clear and had caused the reviewer's concerns.

Some of the reviewer's requests, such as e.g. the wish for mentioning ECA&D and E-OBS earlier than in line 39 and more prominently as well as include those in product/method comparison seemed arbitrary and at times hints a conflict of interest.

Nevertheless, we have put great effort into the rewriting of the introduction to improve the scene setting as the reviewer calls it to assure that the reason, distinction and use of EMO-5 is crystal clear.

[revised manuscript text omitted]

**2. The reason why the analysis is limited to 1970 needs to be made far more explicit. There are many data that extend back further than 1970 and the choice of 1970 is arbitrary with no clear rationale given to justify the choice. From an application perspective there would be considerable value to extending the analysis to earlier periods to the extent possible.**
**Given that the product is reputedly available from 1970 onwards it is problematic that Figures 1-3 only show availability post-1990. This gives the impression that you may be wishing to hide / obscure availability of data in the first 20 years of the series. The figures should show 1970 onwards or the product should be cut at 1990 and published as such. Later at the end of section 3 you suggest this is the case. Please clarify whether the product extends 1970 to date or 1990 to date and ensure consistently stating so throughout the paper.**

Reply to point 2:

The frame in which the meteorological data are collected is described in the introduction (Section 1, lines 32-54): to serve the need of the CEMS, and in particular EFAS. EFAS requires a good coverage of observations for the most recent decades (to calibrate and validate our hydrological model) as well as real time data (for the initialisation of the hydrological model used for operational forecasting). For this reason, we are not collecting earlier data (before 1970). Also, earlier data is already covered by C3S, and hence would be a duplication within the Copernicus service.

With regards to the second paragraph of your comment: In line 48 it is specified that the MDCC is collecting meteorological data from 1970 onwards. This is specifying the time frame of the collection of the in situ meteorological data under CEMS, but not the starting time of the product (EMO-5), which is explicitly mentioned throughout the manuscript as 1990, see:
- abstract (lines 23-24): *"EMO-5 (release 1) covers the time period from 1990 to 2019,[…]"*
- introduction (line 65): *"[…] (from 1 January 1990)."*
- Section 2 (line 123): *"For each of the seven EMO-5 meteorological variables, the location (and hence density) of the input data, as well as the record length per station and the number of input stations over time (1990-2019),"*
- Figure 1-3
- Section 3.3.6 (lines 322-323): *"The EMO-5 version which will be available on the JRC Data Catalogue has been produced for the whole length of the archive from 1990 till 2019, and it is foreseen to extend this dataset on a monthly base."*
- Section 6 (line 467): *"EMO-5 covers the time period from 1990 onwards […]"*

Despite the consistent reporting of the starting time of EMO-5, we recognise that the sentence in Section 3.3.6 (lines 322-323) and in particular *"for the whole length of the archive"* might have caused the confusion, for which reason we modified it to: *"The EMO-5 (version 1) dataset has been produced for the time period from 1990 till 2019."*

**3. The methodology in almost all aspects needs to be strengthened and much more explicit. The method must be outlined to the extent that the analysis could be to first order independently reproduced based upon the description. This is necessary and a central tenet of the scientific method itself. Please revise the method to be considerably more comprehensive such that based upon the method description an independent researcher may reasonably be able to recreate your approach.**

Reply to point 3:

We have checked this point, but we fail to see where the methodology is not described in enough detail to be reproducible. Obviously the input data are not provided and neither the code, but beside those we have described all that is needed to reproduce the methodology including the variable-specific data validation rules, the interpolation algorithm, the station selection criteria, the variable specific aggregation periods and any other variable specific information that is needed.

It has also not been a comment of the other two reviewers, so it is difficult for us to understand where an expansion would be required in the opinion of the reviewer.

**4. The validation activity would be stronger if it could be shown in more detail for all variables and would, undoubtedly, serve to increase user uptake. However, the only real validation in the strict scientific sense is actually the leave-one-out analysis which is not even contained within the validation section at all. What is presently classed as being validation is actually characterisation of the product. Validation requires a value that is of known quality to compare against and none of the three sub-sections in the section purportedly to do with validation actually undertake such a comparison. That section should be retitled to Characterisation accordingly. Then the leave-one-out validation should be considerably expanded, shown for numerous parameters, and aggregated across well-sampled and sparsely-sampled regions separately.**

Reply to point 4:

The leave-one-out validation for all variables is in Table 4 and is explicitly mentioned in line 159. It is referenced in Section 3.2 in which we determine the optimal interpolation scheme for this operational, real-time data set. We have considered moving it under the validation section, but as the choice of the interpolation scheme is part of the methodology we see benefits for leaving it in Section 3.2.

As it wasn't obvious that Table 3 contained the leave-one-out analysis we extended the table caption to: *"Summary of the error measures for the three interpolation schemes based on the leave one out analysis. (Best values are in bold)"*

We do not have a section called "Validation", only called "Evaluation" in which we evaluate the product from various angles, hence we don't think renaming is necessary.

**5. In section 2 the input data is outlined to a very perfunctory level – greater detail here would help including a breakdown by source, data policy etc and links to sources where available publicly by elevating Table A1 to the main text and augmenting with additional information. Also, the maps in Figures 1 to 3 conflate station and gridded input in ways that are really inaccessible. Maps of just primary station source by time and / or variable would be more accessible and avoid proverbially mixing apples and oranges together.**

Reply to point 5:

As the reviewer pointed out correctly, specifications of the input data for EMO-5 can be found in Section 2 and in Appendix 1. The reviewer can find the requested breakdown of sources in Table A1 (line 705), which shows not only the full list of data providers (as requested by the reviewer), but also specifies the variables and the number of stations being collected from each data provider. The table does not contain web links to the data provider as we do not consider it a good practise, as for ones those web links outdate quickly and each data provider can easily be found through using any search engine. Also, CEMS MDCC does not use weblinks for collecting those data, as those are not scraped from

the web, but rather provided directly to us by the data providers listed in Table A1. As the reviewer suggested under point 15 to transfer multi-page figures in the Appendix for their length, we believe that with the same reasoning also Table A1 as a multi-page table should remain in the Appendix. However, if there is a strong preference by the reviewer we have no problems with transferring it to the main text.

We have added information on the data policy at the end of Section 2: *"All data collected by the MDCC, are covered by the EEA-EUMETNET Public Duty License Agreement or the EEA non-EUMETNET Partner license Agreement, which means those data are share-able between the Copernicus services."*

To Figure 1-3: As specified in the Figure captions and in the main text (lines 121ff), they show for each of the EMO-5 variable, the station location, record length per station and the number of stations over time. It is a common practise to show multiple data (sources) in one image as we have done. The differentiation between those was done by colour coding the different data sources, hence there is no mixing of apple and pears.

**Worse, Section 2 is also completely silent on sharing of these station data onwards to e.g. the World Data Centre for Meteorology at NOAA NCEI or the C3S in-situ database effort or ECA&D and thus E-OBS. This limits the potential utility of this data collection activity if you are not actively trying to share data where you can with the activities being undertaken to improve access globally to meteorological holdings, many of which are funded by Copernicus. Where is the joined up thinking? How does this serve Copernicus if you are not actively sharing across Copernicus Services?**

As CEMS is not the owner of those in situ station data we do not have the mandate to distribute them further, with the only exception to other Copernicus services, as the data has been collected under the EUMETNET data licence agreement, which permits that. On that note, we can assure the reviewer that there is an open and strong collaboration between CEMS and the other Copernicus core services, and in particular with C3S to streamline efforts and to avoid duplications. There have been multiple meetings between CEMS-C3S on this and related topics in the past and are pleased to share also with the reviewer that we do support the sharing of those data within the Copernicus services. We do not need to add this information to the manuscript, as the fact that those data are collected under the EUMETNET data licence agreement, does tell the relevant parties (if they don't already know) that those data is sharable within the Copernicus Services.

**6. The quality control is a little limited, in particular in not including any form of neighbour buddy checking as is state of the art in e.g. GHCND or HadISD. Either that or, between the text and the table, the presence of buddy checks is obscured. You should at a minimum be explicit that the applied checks consist of a minimum set of record-based, logic and spike-based checks and do not include a number of other checks including repeat strings, distributional checks, frequent value checks and neighbour-based checks as is the case in state-of-the-art QC procedures such as GHCND or HadISD. You should aim to incorporate a broader array of such checks in future.**

Reply to point 6:

All quality control checks that are currently performed by the CEMS MDCC on the operationally collected meteorological data (and hence also EMO-5) are explicitly mentioned in the "Quality control on input data" section (Section 3.1). We can confirm to the reviewer that there have been no additional checks obscured.

As also to the knowledge of the reviewer, an expansion of the quality control framework of the MDCC has been foreseen and is currently under development, and will be part of the next version of EMO-5. In fact, this is mentioned in Section 6 under "Conclusions and future work" (see lines 500ff): "[…] to improve the current quality control framework, new data validation rules, such as spatial comparison with neighbouring stations or additional statistical checks, will be implemented."

The reviewer has to remember that EMO-5 is created in an operational service to serve a particular purpose. Due to the potential of this data set, we have decided to make is publically available as we see it our duty given that it is a Copernicus product and hence shall be free and open. The QC might be in the opinion of the reviewer limited, but it shall not be forgotten that this data set has been used very successfully for the past decade for the calibration and initialisation of the hydrological model underlying the European Flood Awareness System.

**Furthermore, I would expect the discussion of QC in section 3.1 to include some summary of the frequency with which different values are flagged and some consideration of heterogeneity of flagging of values across sources and regionally. Does the frequency of QC test failure look reasonable? Does it raise any flags for particular sources etc. etc. are all questions I would expect this paper to address if it is to build user confidence in applicability of the resulting product and yet it is presently silent on these issues.**

This is indeed something of interest to the reader. CEMS MDCC publishes every year an analysis on the data collection containing more in detail information, an excellent source for the potential EMO-5 user to receive further information on data providers and provision, database, post-processing, improvements to the data flow and post-processing and gap analysis. We will make sure that next years report contains also statistics on the QC flagging. We have added this source of information in the Conclusion section:

*"Lastly, the CEMS Meteorological Data Collection Centre publishes every year an annual report on the CEMS meteorological data collection, with updated information on e.g. data providers and provision, database, post-processing, improvements to the data flow and post-processing and gap analysis. All reports can be found on the EFAS website, while last years report is referenced here: Rehfeldt, et al., 2021."*

**7. The interpolation scheme discussion in 3.2 requires substantive additional detail. No discussion is forthcoming around why the different schemes have different skill per parameter. This must, intrinsically, be to do with the spatio-temporal correlation**

structures of the different parameters and how effectively the three schemes can handle these. In particular the parameters that vary more smoothly in space and time clearly are better suited to IDW whereas the much smaller scale correlation structure precipitation is better suited to SPHEREMAP. The lack of geophysical interpretation of the findings does not build the necessary user confidence that you understand what underlies your methods.

The analysis should also remark upon the similarity in skill measures for most variables and diagnostics as this implies that interpolation choice isn't a first order effect presumably? Furthermore, is there any gradation in skill between well sampled and sparsely sampled regions? One would presume so but your present aggregation just lumps all cases together. It would be considerably more informative to split this analysis and repeat for well-sampled and sparsely-sampled regions. I would expect no impact of choice in well-sampled regions where interpolation choice makes little difference but much larger impacts in sparsely sampled regions. Can you repeat that analysis but splitting out by regions of greater and lesser station density? What does that tell you?

Reply to point 7:

This is a very interesting comment and these are all valid. But the aim of the comparison was to find the one optimal interpolation scheme for all parameters in terms of computational effort, computational time and lowest uncertainty. As precipitation is of highest importance in hydrologic modelling, a higher weight was given to the performance in precipitation than the other parameters. It was driven by the application side and not to characterize interpolation schemes.
The proposed comparison of the interpolation schemes considering the station density is an additional study and beyond the scope of this paper. A similar study was presented by GPCC in a paper from 2013 at the EGU General Assembly only for precipitation. Briefly, the interpolation uncertainty decreases with increasing station availability. The interpolation schemes didn't depict the same sensitivity, a scheme with the 'lowest' uncertainty at low station density had the 'highest' uncertainty at high station density and vice versa.

8. Given that ERA-Interim/Land values may be biased in terms of variance and mean state isn't the time-varying use of this product to infill in data sparse regions potentially problematic for long-term product homogeneity? Also, ERA5-Land has replaced ERA-Interim -land having higher resolution and improved quality. Should the algorithm not use this product instead? In particular because ERA-Interim land is no longer produced in CDAS mode introducing potentially a major discontinuity into your product upon cessation of ERA-Interim land production?

Reply to point 8:

We do agree with the reviewer on his/her view on the quality of ERA-Interim/Land and ERA5-Land. As specified in Table 1 (line 665) the import of ERA-Interim/Land data as virtual stations has been limited to peripheral areas for which the coverage with in situ stations in

the MDCC database is very low, such as Iceland, North Africa and the eastern part of the EFAS domain (Near East, Caucasus, Russia); visible in Figure 1 (top, left panel). After the discontinuity of ERA-Interim/Land an import of ERA5/Land had been discussed intensively, with the final decision of not including ERA-5/Land. Reason for that decision was that since the start of importing ERA-Interim/Land for data sparse areas more and more stations in those areas were added to the MDCC data base, which diminished the potential gain of adding ERA-5/Land.

With regards to the product homogeneity concern: As specified in the introduction (first and second paragraph) the prime purpose behind the creation of the meteorological grids is to serve the European Flood Awareness System. EFAS requires a good coverage of observations for the most recent decades (to calibrate and validate our hydrological model) as well as real time data (for the initialisation of the hydrological model used for operational forecasting). The more accurate the meteorological data, the better for the application, hence it is not the homogeneity over time that is prime concern, but to have the best possible representation of the meteorological situation for each time step.
We are aware that this requirement differs from applications that focus on climate research. In order to give that credit, we added it as limitation in the Introduction as well as in the Conclusion section:

In the introduction:

*"[…]. Based on the information provided in this paper, the scientific community will understand that EMO-5 (version 1 as well as the operational grids) is not a climatological dataset (unlike e.g. the Essential Climate variables produced by C3S, see Noone et al., 2021), but an operational data set based on the maximum amount of quality-controlled information available at any given time. The implications of this are, that we do not advice to use this dataset to determine climate variability and climate change, however any other environmental application, especially those with real-time, high spatial resolution or multi-variable needs is likely to benefit from this data set. […]"*

In the Conclusion section:

*" The presented EMO-5 (version 1) is the result of an operational real-time service, which utilizes for every grid realisation the maximum amount of valid information available. This is very different to climate data sets, which need to be of sufficient length, consistency, and continuity to determine climate variability and climate change. EMO-5 favours the maximum amount of available information over long-term product homogeneity and is therefore not suitable for climate studies i.e. trend analysis. However, EMO-5 is suitable for environmental application which do not have this homogeneity requirement, but value a data set that uses the maximum amount of information available for each grid realisation."*

**9. The assumptions around 06-18 and 18-06 for Tn and Tx will miss many true Tx and Tn values particularly in the winter half season when daily maxima or minima often occur outside these windows. Particularly so in the higher latitudes of the domain. They also risk introducing a discontinuity with the mean temperatures calculated**

**over the full 24 hours. At a minimum caveats need to be added but, ideally, you would calculate Tx, Tn and Tm consistently for full 24 hour periods to enable cases where Tx and / or Tn fall outside of nominal day / night.**

We assume that the reviewer is familiar with the WMO guidelines on the calculation of daily minimum and maximum temperature (WMO-No. 306) and hence recognise that we are following the WMO standard, which we regard as good practise, especially considering that we are an operational service.

Nevertheless, we are aware about the mentioned implications and have added a paragraph in the conclusions on this topic:

*"Further, there might be some reservations towards using the EMO-5 minimum night-time and maximum day-time temperature as the daily minimum and maximum temperature respectively. To recall, the daily minimum and maximum temperature within EMO-5 are calculated following the WMO guideline (WMO-No. 306), which assumes the minimum temperature to occur between 18:00 and 06:00, and the maximum temperature between 06:00 and 18:00. However, particularly in the winter half season and in the higher latitudes, the daily maxima or minima temperature occur sometimes outside these windows. For this reason, some data sets, such as E-OBS, calculate the daily minimum and maximum temperature over the full 24 hour period. Lavaysse et al. (2018) who investigated the minimum and maximum temperature of EMO-5 (referred to as LisFlood in their paper) and E-OBS with regards to their suitability in the frame of temperature extremes in Europe (heat- and cold waves), came to the conclusion that the two observational datasets showed only minor differences in heat and cold waves occurrences and intensities, which according to Lavaysse is probably due to the good agreement in representing both, the minimum and maximum temperature."*

**10. More details are required upon the land-sea mask used and how it is aggregated to the 5km grid. Are stations omitted based upon whether land or ocean at the native e.g. 5 minute resolution of the mask but then the gridded product produced for all 5km gridcells that contain any land? The method is not reproducible or understandable absent of such details.**

Reply to point 10:

Thank you, this paragraph could indeed be misunderstood. The decision about the inclusion or exclusion of a station from gridding is done independently from the land-sea-mask. Per default, all stations within the EFAS domain (European window, plus neighbouring countries) are used for gridding, even if these are at very small islands or drilling platforms. If a station reports very irregular, low quality or no data, then this station is excluded from gridding. It is now reworded as:

*"A land-sea mask is used to exclude sea surfaces from the gridding procedure, as EMO-5 originates from the need for near real-time information on observed meteorological conditions over land surface areas. The land-sea-mask is not used to define stations to be excluded from gridding."*

**11. Section 3.3.4 (Implications of altitude for temperature and water vapour pressure) is a little basic compared to state-of-the-art spline fitting techniques as used e.g. in HadEX or CRUTS. Why has such a relatively speaking simple approach been used as is described in 3.3.4 and how does it compare to spline fitting or other techniques?**

Reply to point 11:

EMO-5 is an operational, real-time data set. Hence the entire methodology needs to be robust and executable in a couple of hours. We tested some techniques to fit parameters for the interpolation to the available data for the actual time stamp, but the quality of the fitted parameters was often below well-developed climatological means. Sometimes the fitting routine even didn't converged.

**12. The mean temperature discussion in 3.3.5 would benefit from discussing various pieces of literature about how to create mean temperatures and the difference between the mean of a 24 hour period, the mean of max / min and various other alternative methods. There have been several papers on the different ways to calculate means and the random and systematic effects they can have. For the purpose of your dataset the key issue is whether the different ways you have done so may impart systematic or random effects. This can be ascertained from a representative (climatologically) set of well-sampled stations being deliberately degraded to calculate the mean from the 24(+) instantaneous values and then various approximations. This can then at least bound your uncertainties.**

Reply to point 12:

We have added a reference to Luedeling (2018), who interpolated hourly time series from min-max values. We have also done a simple cross-validation of a set of simulated 6-hourly times series and compared with the observations, both for the entire time series, and by looking at the 6-hourly averages for the different times of the day separately. We agree that both the methodology and the data set would make it possible to extend both analyses and the discussion on this topic considerably, but we see this as out of scope for this manuscript due to length issues.

We have extended Section 3.3.5 as follows:

*"The goodness of this approximation was analysed in a simple cross-validation procedure. We picked around 800 stations that already had 6-hourly observations from the start of the period (1990). For these stations, we compared the 6-hourly observations with the simulated 6-hourly dataset based on maximum and minimum temperatures. A hexbin-plot is shown in Figure 7, where darker colors indicate a high density of points indicate a temperature with a large number of observed and simulated values. The line shows where simulations are equal to the observations, whereas the dashed lines indicate where the simulated values are 2 degrees higher or lower than the observations. We can see that a large majority of the*

*simulated values are within 2 degrees of the observations, although there are cases with larger deviations.*

*The correlation between observations and simulations range from 0.91-0.99. The simulated values are on the average unbiased, with mean and median around 0.1 degrees below the observations. However, when looking at the simulated temperatures for different times of the day, it can be noted that the method has a tendency to underestimate the average night temperature, and overestimate the afternoon temperature. The mean and median of the underestimation is 1.8 degrees for the night temperature and the overestimation 1.5 degrees for the afternoon, i.e., the simulated 6-hour periods are slightly more extreme than the observations. The root mean squared error (RMSE) for each station ranged from 1-4 degrees, with mean and median RMSE of 2.2 degrees. The difference was seen as acceptable for our purposes, and is quite similar to what was observed by e.g. Luedeling (2018)."*

**13. I would expect considerably more in the data availability section. It is grossly insufficient to just point to a website. You must describe things such as the data format(s) (ideally there should be at least two to cater to a range of users), any software tools available, version control, version retention policies, whether files are available as spatial or temporal aggregations, whether the files contain uncertainty information, whether they are univariate or multivariate. Otherwise no user is going to go to that site on the off chance. You are not really helping yourselves to advertise the availability if all you do is point to a URL. Users need to know what they are being offered and how well it is ultimately managed.**

Reply to point 13:

We followed the official instructions for submission provided by ESSD with regards to what should be in the individual sections (see https://www.earth-system-science-data.net/submission.html). For the data availability section it reads as follows:

> Data availability: this section should contain all necessary information on data access. Upon acceptance for publication, this includes at least the data set DOI(s) and their data-citation(s). If more than 5 DOIs are necessary, please create a table including DOIs and citations and reference this table here instead. This is also the section that holds review links or other access tokens to your data set if necessary for the review. Optionally, overarching project URLs or other meta-information can be included. The reader must gain direct access to the data by the means described in this section.

Despite being regarded as "grossly insufficient" by the reviewer, the data availability section is in line with the journal's content guideline. Nevertheless, we agree with the reviewer on the value of that information for which reason we have extended the data availability section, and have created a README.txt which has been uploaded to the data repository.

*4 Data availability*
*EMO-5 (Thiemig et al., 2021) is a Copernicus product and as such free and open to everyone. It can be accessed through the Data Catalogue of the European Commission's Joint Research Centre at https://doi.org/10.2905/0BD84BE4-CEC8-4180-97A6-8B3ADAAC4D26.*

*The repository contains a CF-1.6 compliant NetCDF stack files for each variable, as well as a README file with detailed product specifications, which are briefly summarised in Table 5 below.*

**Table 5: Product specification of EMO-5.**

| Data set description | |
| --- | --- |
| **Name of the data set** | *EMO-5 (EMO stands for "European Meteorological Observations", whereas the 5 denotes the spatial resolution of 5 km)* |
| **Short description** | *EMO-5 (version 1) is a European high-resolution, (sub-)daily, multi-variable meteorological data set built on historical and real-time observations obtained by integrating data from 18,964 ground weather stations, four high-resolution regional observational grids (i.e. CombiPrecip, ZAMG - INCA, EURO4M-APGD and CarpatClim) as well as one global reanalysis (ERA-Interim/Land). EMO-5 includes at daily resolution: total precipitation, temperatures (minimum and maximum), wind speed, solar radiation and water vapour pressure. In addition, EMO-5 also makes available 6-hourly precipitation and mean temperature. The raw observations from the ground weather stations underwent a set of quality controls, before SPHEREMAP and Yamamoto interpolation methods were applied in order to estimate for each 5x5 km grid cell the variable value and its affiliated uncertainty, respectively.* |
| **Created by** | *Copernicus Emergency Management Service* |
| **Horizontal coverage** | *Europe (EFAS domain)* |
| **Horizontal resolution** | *5 km* |
| **Spatial gaps** | *Only land areas are covered by this data set* |
| **Temporal coverage** | *1990-01-01 till 2019-12-31* |
| **Temporal resolution** | *daily and 6-hourly* |
| **Temporal gaps** | *No gaps* |
| **Number of available variables** | *7* |
| **Variables available at daily resolution** | *total precipitation, temperatures (minimum and maximum), wind speed, solar radiation and water vapour pressure* |
| **Variables available at 6-hourly resolution** | *total precipitation and mean temperatures* |
| **Units** | *precipitation [mm], temperature [°C], vapor pressure [hPa], solar* |

| | |
|---|---|
| | *radiation [J/m²], wind speed [m/s]* |
| *Projection* | *Lambert Azimuthal Equal-Area (5km)* |
| *Data type* | *5\*5 km grids* |
| *Available version* | *version 1* |
| *DOI of dataset* | *https://doi.org/10.2905/0BD84BE4-CEC8-4180-97A6-8B3ADAAC4D26* |
| *PID of dataset* | *http://data.europa.eu/89h/0bd84be4-cec8-4180-97a6-8b3adaac4d26* |

**14. seNorge2 and seNorge2018 are as I understand it consecutive versions of a single dataset with 2018 replacing 2. You should use just the 2018 version as the other version has been deprecated. Having also reviewed that paper and in particular considering their discussion it would furthermore not be appropriate to treat that product as truth given the substantial caveats made around interpolation in data sparse areas of complex topography. Therefore the seNorge2018 validation must be noted to be conditional on the verity of that product which cannot be assured a priori. You cannot treat either seNorge version as 'truth' which significantly inhibits their value as a means of validation. There are many other national gridded products and comparison to a range of such products e.g. the UK analysis from the Met Office may be instructive and reduce the dependency upon a single product which may itself contain substantive systematic biases.**

Reply to point 14:

In the absence of knowing the actual "truth" as neither in situ measurements nor high-resolution do represent the truth under all circumstances, we do not treat either of those as truth, and have neither suggested it in the manuscript. What we did however do is an intercomparison, and we do learn from those through identifying similarities and differences in the products, especially if we do understand their underlying production methodologies. That said, we do find value in not only comparing it to seNorge2018, but also to seNorge2, as the newer dataset is not a simple successor. Apart from the number of stations in seNorge2018, there were very large differences and evolution in the methodology used to create it, which are very well documented in the literature (referred to as Lussana et al., 2018 and Lussana, 2018b in the paper). The use of both datasets has the advantage to provide ideas on how to improve the EMO-5 dataset in the future and what influence the different approaches have (topography, use of reanalyses, ...). The changes and limitations are very well documented, which is very helpful for the interpretation of the results. For this very reason, the authors consider it useful to include both data sets.

Of course, a comparison with many different high-resolution regional datasets would be interesting. However considering already the length of the manuscript out of scope. At the same time, the comparison done with seNorge data is without doubt interesting and useful,

as the seNorge datasets as an example fulfils many considerations: high-resolution datasets, which is very important for a high-resolution dataset like EMO-5 as a comparison; the Norwegian topography is very challenging; there are data sparse and dense areas; and it's documentation. In addition, as described above, there are versions with very different methodologies, which the authors see as a positive. Because this helps to understand the EMO-5 dataset with both strengths and limitations.

**15. I cannot make head or tail of Figure 11 from the caption and the individual images. The caption needs expanding and the panels given better titles. I cannot understand why the panels in each case vary in the manner they do and it is non-intuitive to me and therefore will be so to your readers. A figure extending over several pages is also barely legible. You would probably be better to concentrate upon 1 or 2 events that could be given more space in the main text and place the remaining cases in supplemental materials.**

Reply to point 15:

We do agree that our analysis of extreme precipitation events is quite extended with a long figure. However, knowing that we are interested in using the data set for reproducing floods the capacity of the data set to capture high precipitation events is of particular interest/ importance to us, and hence worth investigating. Also, we have included numerous events as we did not want to choose a well-working one and then call it a 'representative' example. We believe this has a much higher value to the reader than 1-2 cherry-picked examples.

**16. Lacking in entirety is any meaningful discussion section which may highlight caveats, future work avenues, future priorities, synergies with other Copernicus and global activities (such as WMO) that may yield improvements etc. etc. I would expect a substantive discussion section which covered such points.**

Reply to point 16:

Future plans for EMO-5 are mentioned under Section "6 Conclusions and future work", see last paragraph (l 498ff), which reads as follows:

*"As part of the operational Copernicus EMS, the number of stations (historical and near real-time) that are used for gridding in EMO-5 will be continuously increased, through adding new data providers and the integration of new, high-resolution regional observational grids, where available. In addition, to improve the current quality control framework, new data validation rules, such as spatial comparison with neighbouring stations or additional statistical checks, will be implemented. Finally, as it is foreseen to increase the spatial resolution of EFAS from the current 5 km grid to a 1 arc minute grid (approximately equal to 1.8 km at the equator), also EMO-5 will increase the spatial resolution in its next version."*

Synergies with other Copernicus services is a continuous endeavour and takes place not through discussing/stating those in scientific publications, but via direct dialogues and collaborations, which we practice both.

As already mentioned under point 8, we have added a paragraph to the conclusion section which highlights caveats and limitations of EMO-5. It reads as follows:

*"The presented EMO-5 (version 1) is the result of an operational real-time service, which utilizes for every grid realisation the maximum amount of valid information available. This is very different to climate data sets, which need to be of sufficient length, consistency, and continuity to determine climate variability and climate change. EMO-5 favours the maximum amount of available information over long-term product homogeneity and is therefore not suitable for climate studies i.e. trend analysis. However, EMO-5 is suitable for environmental application which do not have this homogeneity requirement, but value a data set that uses the maximum amount of information available for each grid realisation.*

*Further, there might be some reservations towards using the EMO-5 minimum night-time and maximum day-time temperature as the daily minimum and maximum temperature respectively. To recall, the daily minimum and maximum temperature within EMO-5 are calculated following the WMO guideline (WMO-No. 306), which assumes the minimum temperature to occur between 18:00 and 06:00, and the maximum temperature between 06:00 and 18:00. However, particularly in the winter half season and in the higher latitudes, the daily maxima or minima temperature occur sometimes outside these windows. For this reason, some data sets, such as E-OBS, calculate the daily minimum and maximum temperature over the full 24 hour period. Lavaysse et al. (2018) who investigated the minimum and maximum temperature of EMO-5 (referred to as LisFlood in their paper) and E-OBS with regards to their suitability in the frame of temperature extremes in Europe (heat- and cold waves), came to the conclusion that the two observational datasets showed only minor differences in heat and cold waves occurrences and intensities, which according to Lavaysse is probably due to the good agreement in representing both, the minimum and maximum temperature."*

---

## Referee Report (RR1)

**Review of 'EMO-5: A high-resolution multi-variable gridded meteorological data set ffor Europe' by Vera Thiemig et al.**

This is a second review of this paper following the open public review period. The authors have addressed some, but not all, comments provided by myself and the other reviewer. The work undertaken to address these review comments has undoubtedly served to improve the paper and increase potential uptake of the product by users. There remain a number of points that I believe require to be addressed before this can be published to enable the analysis to be understandable to the ESSD readership and to enable reproducibility of the results.

**Major**

1. Lines 89-96 have now arguably gone too far the other way. You probably want to say something like: 'Users should be aware that EMO-5 is prepared principally for near real-time rather than climatological applications. While the series are available from 1990 users intending to apply the dataset for climatological applications should take care in its application and consider, in addition, the use of other products to ensure the robustness of their analysis to the choice of dataset. For the station database aspects of the product users may also consider the C3S holdings (Noone et al., 2021) who undertake an expanded suite of delayed mode quality checks and with whom we are actively collaborating regarding the sharing of data sources where the data licensing permits to improve both products. For gridded data products users may consider the E-OBS product and various flavours of global and regional reanalysis served via the C3S climate data store in addition to EMO-5 to assure themselves of the quality of the various products and the robustness of their analyses.' This or similar text would appropriately caveat without ruling out the potential use of EMO-5.

2. I leave this to the editor to determine but my view remains that the list of providers table should be in the main text and not the appendix and that the text introducing them in lines 104-106 should be somewhat expanded.

3. The methodology to my view remains insufficient to enable reproducibility of results. I note that the authors were unclear what I meant here. It is not that the steps aren't present but rather that too many of the steps are described to a perfunctory level and without settings / parameter values given and that this would preclude a reasonable effort at independent replication. For example, the enumerated list line 156-161 solely hints at what was done for some aspects which the table does not cover. E.g. what are the monthly statistics check? Some kind of climatological check? If so what period of climatology and what are the thresholds? Note that I am using this as an example only. I would urge the authors to carefully reread the methodology and consider whether each and every step is described in sufficient detail that a reader might be able to reasonably approximate their method based upon the description given alone. Too often I am left feeling that there is grossly insufficient detail to enable a reasonable attempt at being able to reconstruct the method. Given that reproducibility is the central tenet of the scientific process I would urge the authors to carefully redraft their methodology providing sufficient details as to approaches and specific parameter choices to enable an independent replication.

4. **Table 3 could be expanded to provide the precise parameter settings used in your approach.** This is presently given solely for Modified SPHEREMAP but presumably both remaining approaches also had to have some of the parameters set to give values. This comment is by way of a further example whereby the method reproducibility is questionable.

5. You need to at least briefly describe what the Yamamoto method for uncertainty quantification is and if there were parameters that needed to be given a value the values you chose should be given. Hence there is a need to revisit the paragraph starting line 206, again with a view to method reproducibility.

6. In the paragraph starting line 251 I think identical stations is perhaps a misleading term. It's a single station but it has redundant records arising from two or more distinct sources so this isn't a case of two or more stations but rather two or more copies of the records from a given station. I think it would be better to talk about redundant versions of records from some stations that have been shared multiple times and that you make steps to identify such records and mingle them to produce a single record for any given station. I would suggest a rewrite of this paragraph accordingly so that it is clearer to a reader what is going on here and use redundant records rather than identical stations as the term in particular to be much clearer what is the issue.

7. The data availability is still to me an issue, although the authors are thanked for making some efforts in this direction. Specifically it helps to have specified the file type. The point about meeting journal minimum requirements is noted. But I assume that the authors wish their data to be used by the broadest possible audience and therefore they should aspire to more than simply treating it as a journal tick box exercise. I am missing here details that may really help a reader to have confidence in the data. So, I would retitle section 4 to be "Data availability, versioning and user support". What is there is good but needs augmenting with, for example:
    a. At what delay are various products made available and how are users alerted to e.g. period of record updates
    b. What version control exists, if any?
    c. What user support functionality exists?
    d. Where and how are data issues and notices handled

    This section really should be building the confidence in the user that this is a well documented and well maintained database that they can rely on with confidence. At present its not quite there. The section also may benefit from moving later in the order to come just prior to the conclusions.

8. As the authors note in their responses to the initial review in section 5 they are characterising the dataset. I would therefore be more comfortable with section 5 if it were titled 'characterisation' or 'product characterisation'. Evaluation has implications – at least to a native English speaker - that inferences are being made about the correctness or verity of the product. As noted in my initial review the very nature of the problem precludes such an assessment, sadly. Similarly I would change the opening paragraph of this section accordingly.

9. The text on lines 527-529 would need revision to account for the comment above.

10. Figures 1-3 remain an issue for me in that too many of the details are simply impossible to discern and the use of multiple different symbols is really hard to untangle. Much of the key text which might help to disentangle and understand the

figures is so small as to be indecipherable without zooming in. Considerable efforts are required to make figures 1-3 more user friendly and, in particular, please ensure all text and numbers are readable at the intended final figure size as readers who print it off shall not have the luxury of being able to zoom.

11. Figures 8 and 9 please make the font sizes in these figures larger so they can be read. The keys are impossible to read even scaled to 200% resolution so would be entirely indecipherable for a reader of a printed copy.

12. Figure 12 is somewhat improved but still to me very hard to decipher. I find the colour scheme non-intuitive. I'd expect heavier precipitation amounts to be blue not brown. The colour scheme is also not colour blind friendly

13. Following on from point 12 this colour blind issue actually pertains to all figures. None of the colour schema chosen for figures are colour-blind friendly. A substantial proportion of the global population are colour blind. Several colour blind palettes exist see e.g. https://colorbrewer2.org/ including sequential schema that the authors could choose from. For example figure 12 could use https://colorbrewer2.org/#type=sequential&scheme=BuPu&n=9 with the lightest hues pertaining to the lightest precipitation. This schema would be visible and interpretable to all variations of colour-blindness.

**Minor**

1. The opening paragraph of the introduction feels like it is missing important context. What aspects of the data quality? Is it their absolute quality? Assurance of their quality? Something else? What do you mean by environmental and risk indicators? Perhaps give an example?

2. Line 78 I would suggest 'substantial' rather than 'long'. 1990 is not long in the grand scheme of things from a climatological perspective.

3. There should be a line break after line 88 assuming this is intended to be a new paragraph?

4. Line 111 I think you need to say […]others provided data only for […]

5. Line 184 ECA&D

6. Line 199 I am unclear what you mean by 'given that the resulting grid quality of all variables allows this' – it makes no sense to me, at least in the context in which it is given. Please clarify.

7. Line 246 please be specific which variables or is it all remaining variables in which case say all remaining variables. Also, does this mean that the gridding only considers stations with some minimum set of observed variables and how does this impact station counts etc from sources that have not all variables? Again, this lack of detailed description is precluding reproducibility (see major comments)

8. Line 281 – please specify which land sea mask is used to enable reproducibility

9. Line 290 – which DEM is used? Again, you need to specify to enable reproducibility

10. Line 302 – add 'as follows' to the end of this sentence to be clear that the modifications are then described in the next paragraph. Either that or describe what those modifications were here.

11. Line 315 – presumably the 5.5 has units. What are these? $K^2$?

12. Line 359 if a new paragraph should have a line break. Same at line 365

13. Lines 378-380 the enumeration should match the section ordering that follows
14. On line 430 'two blue patches' is very colloquial. Can more scientifically robust language be used in redrafting please?
15. Line 454-455 was there really somewhere in Norway with no precipitation for 1400 consecutive days or is this some aggregate of this statistic over some region? As written this is really unclear and a redraft is required for clarity here.
16. Line 463 if a new paragraph should have a line break added
17. Line 496 substantial rather than larger (I think)
18. Line 507 station at end of sentence should be stations?
19. In figure 7 could more sensible bin boundaries be used? It feels really odd to use counts ending in random numbers rather than 0, 40,000, 80,000 etc.

---

## Author Response (AR2)

**Review of 'EMO-5: A high-resolution multi-variable gridded meteorological data set for Europe' by Vera Thiemig et al.**

This is a second review of this paper following the open public review period. The authors have addressed some, but not all, comments provided by myself and the other reviewer. The work undertaken to address these review comments has undoubtedly served to improve the paper and increase potential uptake of the product by users. There remain a number of points that I believe require to be addressed before this can be published to enable the analysis to be understandable to the ESSD readership and to enable reproducibility of the results.

Dear reviewer,

thank you for your continuous dedication to improving our manuscript. As in the previous round of review, we have read attentively each and every of your comment and addressed each with care and consideration, so that we can move forward with the full publication of this manuscript.

With best regards,

Vera Thiemig

**Major comment 1:**

Lines 89-96 have now arguably gone too far the other way. You probably want to say something like: 'Users should be aware that EMO-5 is prepared principally for near real-time rather than climatological applications. While the series are available from 1990 users intending to apply the dataset for climatological applications should take care in its application and consider, in addition, the use of other products to ensure the robustness of their analysis to the choice of dataset. For the station database aspects of the product users may also consider the C3S holdings (Noone et al., 2021) who undertake an expanded suite of delayed mode quality checks and with whom we are actively collaborating regarding the sharing of data sources where the data licensing permits to improve both products. For gridded data products users may consider the E-OBS product and various flavours of global and regional reanalysis served via the C3S climate data store in addition to EMO-5 to assure themselves of the quality of the various products and the robustness of their analyses.' This or similar text would appropriately caveat without ruling out the potential use of EMO-5.

Reply by authors:

We have adopted the first suggested sentence into the paragraph to stress right from the beginning that EMO-5 is not targeted towards climatological applications, and used the rest of the paragraph for what EMO-5 can be considered.
As this is a scientific publication on a data set which is clearly stated to not aim at climatological applications, we do not see reason to refer to other data sets in this paragraph. Other existing datasets, among which E-OBS, are credited in a more prominent location three paragraphs earlier (ll 68-80).

Changes in manuscript:

*before:*
Based on the information provided in this paper, the scientific community will understand that EMO-5 (version 1 as well as the operational grids) is not a climatological dataset (unlike e.g. the Essential Climate variables produced by C3S, see Noone et al., 2021), but an operational data set based on the maximum amount of quality-controlled information available at any given time. The implications of this are, that we do not advice to use this dataset to determine climate variability and climate change, however any other environmental application, especially those with real-time, high spatial resolution or multi-variable needs is likely to benefit from this data set. Hence, by making the EMO-5 (version 1) data publicly available, we aim to support many other environmental applications and services that would benefit from using those data, such as e.g. hydrological, agricultural or other environmental applications.

*now:*
Users should be aware that EMO-5 is prepared principally for near real-time rather than climatological applications. EMO-5 (version 1 as well as the operational grids) is an operational data set based on the maximum amount of quality-controlled information available at any given time. Environmental applications, especially those with real-time, high spatial resolution or multi-variable needs are likely to benefit from this data set. Hence, by making the EMO-5 (version 1) data publicly available, we aim to support many other environmental applications and services that would benefit from using those data, such as e.g. hydrological, agricultural or other environmental applications.

**Major comment 2:**
I leave this to the editor to determine but my view remains that the list of providers table should be in the main text and not the appendix and that the text introducing them in lines 104-106 should be somewhat expanded.

Reply by authors:

Table A1 contains the listing of data providers that have contributed with their data to EMO-5, hence a very relevant information. Due to its substantial length (>1 page) we found it more appropriate to put it in the Appendix, where it is available to each reader as all the other tables of this manuscript, however, without interrupting the core text.

At this point, we agree for the editor to decide.

**Major comment 3:**
The methodology to my view remains insufficient to enable reproducibility of results. I note that the authors were unclear what I meant here. It is not that the steps aren't present but rather that too many of the steps are described to a perfunctory level and without settings / parameter values given and that this would preclude a reasonable effort at independent replication. For example, the enumerated list line 156-161 solely hints at what was done for some aspects which the table does not cover. E.g. what are the monthly statistics check? Some kind of

climatological check? If so what period of climatology and what are the thresholds? Note that I am using this as an example only. I would urge the authors to carefully reread the methodology and consider whether each and every step is described in sufficient detail that a reader might be able to reasonably approximate their method based upon the description given alone. Too often I am left feeling that there is grossly insufficient detail to enable a reasonable attempt at being able to reconstruct the method. Given that reproducibility is the central tenet of the scientific process I would urge the authors to carefully redraft their methodology providing sufficient details as to approaches and specific parameter choices to enable an independent replication.

Reply by authors:

We have gone through the description of the methodology and added information at numerous places, such as to the availability check, the monthly check and the interpolation methods (including parameterisation).

Changes in manuscript:

*before:*

[revised manuscript text omitted]

• Initial search radius as in Shepard, 1968 |

**Major comment 4:**
Table 3 could be expanded to provide the precise parameter settings used in your approach. This is presently given solely for Modified SPHEREMAP but presumably both remaining approaches also had to have some of the parameters set to give values. This comment is by way of a further example whereby the method reproducibility is questionable.

Reply by authors:

We have added a column to Table 3, providing the information on the parameterisation. (Please see above our reply to major comment 3)

**Major comment 5:**
You need to at least briefly describe what the Yamamoto method for uncertainty quantification is and if there were parameters that needed to be given a value the values you chose should be given. Hence there is a need to revisit the paragraph starting line 206, again with a view to method reproducibility.

Reply by authors:

A description of the method can be found in the paragraph starting at line 206. It also contains the reference to the methodology for those with deep reading wishes. This method does not require a parameterisation, which information we have added to the manuscript.

Changes to the manuscript:

There are many methods available to estimate the uncertainty (i.e. reliability) of the gridded values, such as the leave-one-out approach, ensemble creations or the technique developed by Yamamoto (2000). Kriging itself provides an error estimation, but this depends only on the spatial distribution of the applied stations and not on the input data, therefore this error estimation is not applicable here. As the computational time of the grids is highly relevant in order to produce the operational grids as input for emergency management applications, the technique developed by Yamamoto (2000) is used due to its low computational effort. Furthermore, this method takes into account the variability of the surrounding observations, unlike the common Kriging uncertainty that only depends on the variogram and the spatial distribution of stations. This approach was also used, for example, for the E-OBS data set (Haylock et al., 2008). Originally, it was developed for Kriging schemes, but was adapted to the utilized modified SPHEREMAP scheme. Briefly, the method uses the interpolation weights to calculate a weighted variance between the gridded value and the input station data. It is zero if all input data are identical (e.g. areas with zero precipitation) and increases with increasing variance of the input data. This method does not need any additional information besides grid value, station values and the interpolation weights.

**Major comment 6:**
In the paragraph starting line 251 I think identical stations is perhaps a misleading term. It's a single station but it has redundant records arising from two or more distinct sources so this isn't a case of two or more stations but rather two or more copies of the records from a given station. I think it would be better to talk about redundant versions of records from some stations that have been shared multiple times and that you make steps to identify such records and mingle them to produce a single record for any given station. I would suggest a rewrite of this paragraph accordingly so that it is clearer to a reader what is going on here and use redundant records rather than identical stations as the term in particular to be much clearer what is the issue.

Reply by authors:

We agree to this comment and appreciate the suggestion, which we have implemented throughout the manuscript.

Changes in manuscript:

*before:*
For precipitation, not all stations that fulfil the above criteria are used in the interpolation. This is due to the fact that over time, there was an increasing number of identical stations that were reported by different data providers (e.g. identical stations are often found between the SYNOP as well as national data), albeit sometimes with slightly different values or slightly different coordinates. Not removing those duplicate stations would lead to a multiple counting of the same station during the interpolation, with the result of overweighting of those stations in the grids and less reliable area mean grid-cell values. To correct this, and to assure a gradual change between stations during the interpolation, duplicate stations within a vicinity of 500 metres were identified and merged into one virtual station. The coordinates of the virtual station were taken from the first station of this cluster, while the value was computed as the average of all duplicate stations. This reduced the total number of stations used per grid realisation by an average of 3.4%. Figure 1 shows the number of stations used during the grid creation.

*after:*
For precipitation, not all station records that fulfil the above criteria are used in the interpolation. This is due to the fact that over time, there was an increasing number of stations that were reported redundantly by different data providers (e.g. SYNOP and national data), albeit sometimes with slightly different values or slightly different coordinates. Hence appearing as multiple station in the database, while in fact they are one station with multiple records. Not removing those duplicate records would lead to a multiple counting of the same station during the interpolation, with the result of overweighting of those stations in the grids and less reliable area mean grid-cell values. To correct this, and to assure a gradual change between stations during the interpolation, redundant records were identified through a vicinity check. Records, i.e. stations within a vicinity of 500 metres to each other were identified and merged into one record, i.e. virtual station. The coordinates of the virtual station were taken from the first station of this cluster, while the value was computed as the average of all duplicate records. This reduced the total number of records used per grid realisation by an average of 3.4%. Figure 1 shows the number of stations used during the grid creation.

**Major comment 7:**
The data availability is still to me an issue, although the authors are thanked for making some efforts in this direction. Specifically it helps to have specified the file type. The point about meeting journal minimum requirements is noted. But I assume that the authors wish their data to be used by the broadest possible audience and therefore they should aspire to more than simply treating it as a journal tick box exercise. I am missing here details that may really help a reader to have confidence in the data. So, I would retitle section 4 to be "Data availability, versioning and user support". What is there is good but needs augmenting with, for example:
a. At what delay are various products made available and how are users alerted

to e.g. period of record updates
b. What version control exists, if any?
c. What user support functionality exists?
d. Where and how are data issues and notices handled
This section really should be building the confidence in the user that this is a well documented and well maintained database that they can rely on with confidence. At present its not quite there. The section also may benefit from moving later in the order to come just prior to the conclusions.

Reply by authors:

EMO-5 being a Copernicus product, the general Copernicus user support service will be providing or coordinating user support. In addition, there is a contact person outlined in the JRC Data Catalogue under the link provided in the manuscript.
The version control of EMO-5 will be aligned with the EFAS version control. Documentation on the EFAS version control can be accessed publically on the CEMS wiki:
https://confluence.ecmwf.int/display/COPSRV/EFAS+versioning+system
In addition the readme file of EMO-5 will be kept updated.

**Major comment 8:**
As the authors note in their responses to the initial review in section 5 they are characterising the dataset. I would therefore be more comfortable with section 5 if it were titled 'characterisation' or 'product characterisation'. Evaluation has implications – at least to a native English speaker - that inferences are being made about the correctness or verity of the product. As noted in my initial review the very nature of the problem precludes such an assessment, sadly. Similarly I would change the opening paragraph of this section accordingly.

Reply by authors:

Having consulted multiple native speakers (from UK, Ireland and US) on this issue, we do come to the conclusion that we do not agree on the proposed term "characterisation".
We do evaluate our product from multiple angles through analysis and comparison, and therefore we have been assured by our native speakers that "Evaluation" is indeed the correct term and therefore used as section title.

**Major comment 9:**
The text on lines 527-529 would need revision to account for the comment above.

Reply by authors:

Please see reply to major comment 8 above.

**Major comment 10:**
Figures 1-3 remain an issue for me in that too many of the details are simply impossible to discern and the use of multiple different symbols is really hard to untangle. Much of the key text which might help to disentangle and understand the figures is so small as to be indecipherable without zooming in. Considerable efforts

are required to make figures 1-3 more user friendly and, in particular, please ensure all text and numbers are readable at the intended final figure size as readers who print it off shall not have the luxury of being able to zoom.

Reply by authors:

Justified comment. The Figures have been updated. Originally the figures have been prepared for online publication with 72 ppt and large zoom potential. We have changed the resolution to 300 ppt, which is the standard for printing and set the image width to a maximum of 16 cm, so that it will not get further compressed once implemented into the manuscript. Also all the labels have been enlarged. The original images will still be provided in the supplement, so that people reading the publication online (we presume that are the majority of readers) will still benefit from the high-res images we originally prepared.

Changes in manuscript:

*before and after of Figure 1:*

[Figure]

[Figure]

*before and after of Figure 2:*

[Figure]

[Figure]

*before and after of Figure 3:*

[Figure]

**Major comment 11:**

Figures 8 and 9 please make the font sizes in these figures larger so they can be read. The keys are impossible to read even scaled to 200% resolution so would be entirely indecipherable for a reader of a printed copy.

Reply by authors:

We do agree on this point, and have resolved it in the same manner as major comment 10.

Changes in manuscript:

*before and after of Figure 8:*

[Figure]

[Figure]

*before and after of Figure 9:*

[Figure]

**Major comment 12:**

Figure 12 is somewhat improved but still to me very hard to decipher. I find the colour scheme non-intuitive. I'd expect heavier precipitation amounts to be blue not

brown. The colour scheme is also not colour blind friendly

Reply by authors:

The colour scheme used in Figure 12 is a standard colour scheme of ArcGIS, that has also been used within the PESETA projects without any problems during publication.

Concerning the comment that the colour scheme used might not be optimal for people with colour vision deficiency seems true (we did a colour blind test). This is however a common problem in almost all publications. Below an example of a similar figure of E-OBS, one can see that tailoring to people with colour vision deficiency is not yet a common practice, as all the areas in grey below would not be differentiable for colour vision deficiency. Nevertheless, they have all been published.

Considering that this is not a requirement of this journal (and neither of others), we have not had it in our mind while preparing the manuscript, and due to the workload it would require at this point we restrain from doing so for this time, but we will have it in mind for future work and publications. At the same time, we believe that this is a problem that needs to be addressed at journal level, as it is not solved by individual authors tailoring to the needs for people with colour vision deficiency.

*Colour scheme used by E-OBS*
*(*https://agupubs.onlinelibrary.wiley.com/doi/full/10.1029/2017JD028200*):*

[Figure]

*In grey highlighted the problematic areas for people with colour vision deficiency:*

[Figure]

**Major comment 13:**
Following on from point 12 this colour blind issue actually pertains to all figures. None of the colour schema chosen for figures are colour-blind friendly. A substantial proportion of the global population are colour blind. Several colour blind palettes exist see e.g. https://colorbrewer2.org/ including sequential schema that the authors could choose from. For example figure 12 could use https://colorbrewer2.org/#type=sequential&scheme=BuPu&n=9 with the lightest hues pertaining to the lightest precipitation. This schema would be visible and interpretable to all variations of colour-blindness.

Reply by authors:

See reply to major comment 12.

**Minor comment 1:**
The opening paragraph of the introduction feels like it is missing important context. What aspects of the data quality? Is it their absolute quality? Assurance of their quality? Something else? What do you mean by environmental and risk indicators? Perhaps give an example?

Reply by authors:
We have re-written the opening of the introduction:

Many environmental models rely heavily on the availability of meteorological data. Factors like the accessibility, quality, spatio-temporal coverage as well as spatio-temporal resolution of those meteorological data influence and ultimately determine their modelling capacity. This is further intensified for environmental applications that are running operationally and require quality-controlled, multi-variable meteorological data in near-real time. One prominent example […]

**Minor comment 2:**
Line 78 I would suggest 'substantial' rather than 'long'. 1990 is not long in the grand

scheme of things from a climatological perspective.

Reply by authors:
Done.

**Minor comment 3:**
There should be a line break after line 88 assuming this is intended to be a new paragraph?
Reply by authors:
Done.

**Minor comment 4:**
Line 111 I think you need to say […]others provided data only for […]

Reply by authors:
Done.

**Minor comment 5:**
Line 184 ECA&D

Reply by authors:
Done.

**Minor comment 6:**
Line 199 I am unclear what you mean by 'given that the resulting grid quality of all variables allows this' – it makes no sense to me, at least in the context in which it is given. Please clarify.

Reply by authors:
Resolved. We removed ",given that the resulting grid quality of all variables allows this"

**Minor comment 7:**
Line 246 please be specific which variables or is it all remaining variables in which case say all remaining variables. Also, does this mean that the gridding only considers stations with some minimum set of observed variables and how does this impact station counts etc from sources that have not all variables? Again, this lack of detailed description is precluding reproducibility (see major comments)

Reply by authors:
Yes, this applies to all remaining variables. We changed the sentence to "data coverage for all remaining variables" to clarify this.
It is clearly written that only stations that fulfil this data coverage criterion are used during gridding. Figure 1-3 (right column) shows the correct count of stations used during gridding for each variable.

**Minor comment 8:**

Line 281 – please specify which land sea mask is used to enable reproducibility

Reply by authors:
Our own in-house land-sea mask has been used. With regards to the reproducibility, the method can be reproduced with any land-sea mask. It is clear that not a 100% copy of EMO-5 can be reproduced as this would mean that the raw source data would need to be shared, which the data licence agreement does not permit.

**Minor comment 9:**
Line 290 – which DEM is used? Again, you need to specify to enable reproducibility

Reply by authors:
Reference was added.

Arnal, L., Asp, S.-S., Baugh, C., de Roo, A., Disperati, J., Dottori, F., Garcia, R., Garcia-Padilla, M., Gelati, E., Gomes, G., Kalas, M., Krzeminski, B., Latini, M., Lorini, V., Mazzetti, C., Mikulickova, M., Muraro, D., Prudhomme, C., Rauthe-Schöch, A., Rehfeldt, K., Salamon, P., Schweim, C., Skoien, J.O., Smith, P., Sprokkereef, E., Thiemig, V., Wetterhall, F., Ziese, M.: EFAS upgrade for the extended model domain – technical documentation, EUR 29323 EN, Publications Office of the European Union, Luxembourg, ISBN 978-92-79-92881-9, doi: 10.2760/806324, JRC111610, 2019.

**Minor comment 10:**
Line 302 – add 'as follows' to the end of this sentence to be clear that the modifications are then described in the next paragraph. Either that or describe what those modifications were here.

Reply by authors:
Implemented as suggested.

**Minor comment 11:**
Line 315 – presumably the 5.5 has units. What are these? $K^2$?

Reply by authors:
Implemented as suggested.

**Minor comment 12:**
Line 359 if a new paragraph should have a line break. Same at line 365

Reply by authors:
Implemented as suggested.

**Minor comment 13:**
Lines 378-380 the enumeration should match the section ordering that follows

Reply by authors:

Implemented as suggested.

**Minor comment 14:**
On line 430 'two blue patches' is very colloquial. Can more scientifically robust language be used in redrafting please?

Reply by authors:
Sentence was re-written by our native English speaking colleague to: "However, the presence of two conspicuous blue-coloured areas shows that EMO-5 is characterized by higher precipitation than seNorge2018."

**Minor comment 15:**
Line 454-455 was there really somewhere in Norway with no precipitation for 1400 consecutive days or is this some aggregate of this statistic over some region? As written this is really unclear and a redraft is required for clarity here.

Reply by authors:
We put with this statement a pointer to a weakness of the data set we promote with this paper. We believe that it is very honest to mention this and we should not hide it. For clarification, we have added that these values are unrealistic.

Change to the manuscript:
[…] This concerns for example the high values of the Consecutive Dry Days (CDD) - up to 1400 days without precipitation - in the EMO-5 data set. These unrealistic values can be limited to a small region in the northeast. […]

**Minor comment 16:**
Line 463 if a new paragraph should have a line break added

Reply by authors:
Implemented as suggested.

**Minor comment 17:**
Line 496 substantial rather than larger (I think)

Reply by authors:
Sentence has been modified by our native English-speaking colleague to "For 13 out of the 15 selected events, EMO-5 shows greater precipitation amounts over the event duration […]".

**Minor comment 18:**
Line 507 station at end of sentence should be stations?

Reply by authors:
Implemented as suggested.

**Minor comment 19:**

In figure 7 could more sensible bin boundaries be used? It feels really odd to use counts ending in random numbers rather than 0, 40,000, 80,000 etc.

Reply by authors:

It was not possible to define the values of the buckets with the R package we had originally used. However we have investigated on how to improve it to satisfy the review, please see below for the revised visualisation. Please note that we have added the entire aspect only on the wish of the reviewer, we are still not entirely convinced that this aspect shall even be in the manuscript as it is a bit beyond the scope and blurs the paper in our view, but as we are trying to compromise with the reviewer we have added it to the manuscript.

*previously:*

[Figure]

*now:*

---

## Author Response (AR3)

The authors have responded to the majority of outstanding issues. The paper is substantively approved and in my view suitable for publication after minor amendments. Other issues I had the authors have responded to and I see no value in persisting on these in general and do not wish to be 'that reviewer' in this regard.

**General reply to the reviewer:**

Dear reviewer,

you have re-listed the items of the second round of review in which we had either a different view or were out of scope, but without an explicit request for changes, so we were not sure if you are pressing us to do any further changes or not.

However, given that the status is "accepted after minor review" it seems that we are still expected to change even the last points in which we have justified a different view in order to receive an acceptance for publication from your side.

Hence, we have gone again through the remaining list and tried to satisfy further your comments. Please see below.

Best wishes,

Vera Thiemig on behalf of all co-authors

**Reviewer comment 1:**

I would remain of the view that the table of sources is of sufficient importance that it should reside in the main text and not in an annex and would invite the editor to make a decision on this.

Response from authors:

Even though our view remains that the list of providers table is well placed in the appendix, we have decided to move this table to the main text.

Change to the manuscript:

Moved Table A1 as Table 1, renumbered the remaining tables.

**Reviewer comment 2:**

I remain of the opinion that the data availability section is too short and does not provide the reader with sufficient detail as to the level of user support involved. I think the authors here are too close to their system and are effectively taking for granted that the reader will know the level of user support that is standing behind this product. My fear is that very many of them won't. Would it really hurt to add a paragraph that describes what being in the data catalogue of JRC means in terms of 24/7 user support, up time, help desk, long-term product support, version control, documentation etc? You want users to read this and go "that's a product I can use with certainty that it is well maintained, well supported, operationally available, and therefore I can be certain of being available today, tomorrow, next year and 5 years hence to support my application so i can build my application around it with confidence". I think the authors are doing themselves and their product a very

substantial dis-service not documenting this better. It would only take an additional paragraph but might mean the difference between a reader deciding to use this product or not.

Response from authors:

Based on the previous comments of the reviewer we have added a data availability section (initially it was missing), created a readme file with detailed product specifications and included a brief summary of product specifications directly to the manuscript (see Table 5).

With regards to product support, version control and documentation we have clarified during the last round of review the following:

> *EMO-5 being a Copernicus product, the general Copernicus user support service will be providing or coordinating user support. In addition, there is a contact person outlined in the JRC Data Catalogue under the link provided in the manuscript.*
>
> *The version control of EMO-5 will be aligned with the EFAS version control. Documentation on the EFAS version control can be accessed publically on the CEMS wiki: https://confluence.ecmwf.int/display/COPSRV/EFAS+versioning+system*
>
> *In addition the readme file of EMO-5 will be kept updated.*

To satisfy the reviewer request we have added the above information to the README file.

Change to the README file:

added the following fields:

- D15. Version control: The version control of EMO-5 will be aligned with the EFAS version control. Documentation on the EFAS version control can be accessed publically on the CEMS wiki: https://confluence.ecmwf.int/display/COPSRV/EFAS+versioning+system
- A04. User support service: contact person outlined in the JRC Data Catalogue OR contact the general Copernicus user support service (support@copernicus.eu)

**Reviewer comment 3:**

The reference in line 377 to 2019 is inconsistent with elsewhere where NRT is assured.

Response from authors:

The reviewer confuses EMO-5 version 1 with the operational grids. The difference between those versions are mentioned in a consistent manner throughout the manuscript:

- abstract (lines 23ff): "EMO-5 (version 1) covers the time period from 1990 to 2019, with a near real-time release of the latest gridded observations foreseen with version 2."
- introduction (lines 56ff): "While the service produces daily updated grids (EMO-5 operational grids), we have re-run our archive from 1990-2019 to produce a new long-term dataset, which we refer to as version 1 of the EMO-5."
- input data section (lines 136ff): "For each of the seven EMO-5 meteorological variables, the location (and hence density) of the input data, as well as the record length per station and the number of input stations over time (1990-2019), are shown in Fig. 1 to 3 respectively."
- data creation section (l 376): "The EMO-5 (version 1) dataset has been produced for the time period from 1990 till 2019."

- in README file:
  - D04. Temporal coverage: 1990-01-01 till 2019-12-31
  - D11. Update frequency: not foreseen for version 1

With this clarification, we regard this issue as closed unless the reviewer disagrees.

**Reviewer comment 4:**

I will not insist for this paper but I would very strongly urge for future work moving all plotting etc. software as quickly as possible to use colour-blind friendly schema. This is a pervasive and insidious case of able-ism that disenfranchises a not inconsiderable proportion of the population and for which copious software and tools exist to assure outputs that are not inadvertently discriminatory. I fully accept that there is no intent involved. However, when the tools exist it is behoven on those of us who are not colour blind to duly consider accessibility to our products to those who are

Response from authors:

Replied to extensively during the last round of review. Thank you for regarding this issue as closed for this manuscript.